# Extracellular Polymeric Substances Produced by Actinomycetes of the Genus *Rhodococcus* for Biomedical and Environmental Applications

**DOI:** 10.3390/ijms27010498

**Published:** 2026-01-03

**Authors:** Anastasiia Krivoruchko, Daria Nurieva, Irina Ivshina

**Affiliations:** 1Perm Federal Research Center, 13a Lenin Street, Perm 614990, Russia; daranurieva0@gmail.com (D.N.); ivshina@iegm.ru (I.I.); 2Microbiology and Immunology Department, Perm State National Research University, 15 Bukirev Street, Perm 614990, Russia

**Keywords:** extracellular polymeric substances, actinomycetes of the genus *Rhodococcus*, biomedicine, environmental protection, bioremediation, viscosity properties, biological activities, emulsifying activity, pollutant sorption, flocculating activity

## Abstract

Extracellular polymeric substances (EPSs) produced by actinomycetes of the genus *Rhodococcus* play crucial roles in their ecological success, metabolic versatility, and biotechnological value. This review summarizes existing studies of *Rhodococcus* EPSs, emphasizing the biochemical composition, functional attributes, and practical significance of EPSs, as well as their importance in biomedicine, bioremediation, and other applications (food industry, biomineralization) with respect to the EPS chemical composition and biological roles. *Rhodococcus* species synthesize complex EPSs composed primarily of polysaccharides, proteins and lipids that, like in other bacteria, support cell adhesion, aggregation, biofilm formation, and horizontal gene transfer (and can prevent exogenous DNA binding) and are highly important for resistance against toxicants and dissolution/assimilation of hydrophobic compounds. EPSs produced by different species of *Rhodococcus* exhibit diverse structures (soluble EPSs, loosely bound and tightly bound fractions, capsules, linear and branched chains, amorphous coils, rigid helices, mushroom-like structures, extracellular matrix, and a fibrillar structure with a sheet-like texture), leading to variations in their properties (rheological features, viscosity, flocculation, sorption abilities, compression, DNA binding, and interaction with hydrophobic substrates). Notably, the EPSs exhibit marked emulsifying and flocculating properties, contributing to their recognized role in bioremediation. Furthermore, EPSs possess antiviral, antibiofilm, anti-inflammatory, and anti-proliferating activities and high viscosity, which are valuable in terms of biomedical and food applications. Despite extensive industrial and environmental interest, the molecular regulation, biosynthetic pathways, and structural diversity of *Rhodococcus* EPSs remain insufficiently characterized. Advancing our understanding of these biopolymers could expand new applications in biomedicine, bioremediation, and biotechnology.

## 1. Introduction

Extracellular polymeric substances (EPSs) represent high-molecular weight biosynthetic polymers that are found outside the cell wall, either attached to it or secreted into the extracellular environment. Polysaccharides are considered to constitute a large proportion of EPSs. Therefore, even in some sources, ‘exopolysaccharides’ is a synonym for extracellular polymeric substances [1,2,3]. But in fact, EPSs in addition may consist of extracellular lipids, proteins, nucleic acids, and humic compounds and differ between species [4,5]. In this way, in most research, bacterial EPSs are being purified for further analyses and applications [6,7,8].

The great majority of bacteria live in the forms of cell communities, such as biofilms. EPSs, as a main component of the biofilm matrix, exhibit ecological and physiological functions. They provide an instrument to survive in different conditions by binding xenobiotics, allowing adhesion to substrates and entrapment of nutrients [4,9]. Furthermore, bacterial polysaccharides are multifunctional, valuable compounds. Due to their biodegradability, non-toxicity, water-absorbing capacity, chelating and emulsifying properties, and biological activities, including anti-radiation, anti-fatigue, antioxidant, hypolipidemic, anti-inflammatory, antitumor, and immunomodulatory effects [8,10], EPSs are applied in medicine, pharmaceuticals, cosmetics, the food industry, remediation, agriculture, and biotechnology [11,12].

EPSs are positioned as prospective, universal, renewable sources. Bacterial polysaccharides compared with those from plants and animals and synthetic ones, differ in their numerous valuable qualities. In addition, the production of microbial EPSs is more cost-effective and scalable [13]. They can be produced on a large scale using modern fermentation technology [13], which ensures stable product quality, easy production control, and eliminates concerns over raw material shortages [2].

Actinomycetes of the genus *Rhodococcus* (domain Bacteria, kingdom Bacillati, phylum Actinomycetota, class Actinomycetes, order Mycobacteriales, family Nocardiaceae, https://lpsn.dsmz.de/genus/rhodococcus, last accessed 16 September 2025) are well-known stress-tolerant biodegraders of a wide range of emergent ecopollutants, mainly hydrocarbons and their derivatives. Actinomycetes are known as abundant producers of metabolites, including fatty acids, proteins, amino acids, siderophores, carotenoids, biosurfactants, and polysaccharides [2,14,15]. *Rhodococcus* spp. have unique biosynthetic activities that contribute to their wide persistence, provide them a competitive advantage over other microorganisms, and could be utilized in biotechnologies. The beneficial ability of rhodococci to obtain high value-added products from low-cost substrates, like wastes of pulp, chemical, and pharma industries, livestock wastewaters and sludge, and contaminants, offer the possibility to efficiently recover valuable resources and provide possible waste disposal solutions [16,17,18]. High tolerance of rhodococci to stresses (they keep growing at 4–42 °C, 7–12% NaCl, 6–8 pH, 20–80% solvents, survive starvation and water deficit) [19,20,21], as well as absence of catabolic repression [15], coupled with their ability to use wastes and contaminants as growth substrates, make them more feasible producers in comparison with fast-growing microorganisms like *Escherichia coli* [22]. Specific cases may include less strict conditions of cultivation and environmental applications like wastewater treatment or bioremediation. *Rhodococcus* spp. are considered to be perspective producers of novel bioactive molecules and thus have the potential for application in the framework of sustainable use, highlighting the relevance of expanding the knowledge of their biosynthetic capacities and biotechnology products.

There is no comprehensive investigation about *Rhodococcus* EPSs’ diversity, properties, and capabilities, which brings further difficulties in completely utilizing their full potential. Variations in structures, coupled with potential applications and strain specificity, are supposed to be key factors in developing new products, optimizing extraction methods, and elucidating molecular mechanisms of action. Moreover, optimization of the procedures used for bacteria cultivation and further extracting EPSs guarantees the preservation of its structure and functionality for various applications.

This review is focused on EPS produced by actinomycetes of the genus *Rhodococcus*. The data on chemical composition, biological functions, properties, and applications of rhodococcal EPS are summarized and discussed below.

## 2. Chemical Composition and Its Influence on Activities and Functions of EPSs

There are fractions of microbial EPSs. Slime, capsular, loosely bound, and tightly bound EPSs are distinguished on the basis of the nature of their association with the cells or the extraction method used (Figure 1). It is considered there are two forms of EPSs: bound EPSs (sheaths, capsular polymers, condensed gels, loosely bound polymers, and attached organic materials) and soluble EPSs (soluble macromolecules, colloids, and slimes) [7,23]. In terms of activated sludge used in wastewater treatment, it is important to divide tightly bound EPSs (TB-EPSs) located in the inner layer of the sludge floc and loosely bound EPSs (LB-EPSs) existing in the outer layer of the sludge floc. EPSs comprise a significant part of the biopolymers synthesized by cells. The thickness of EPSs surrounding cells can be similar or even larger than the cell diameter. According to measurements performed with an atomic force microscope, the thickness of TB-EPSs produced by *Rhodococcus* cells varies between 240 nm and 1000 nm depending on the density of the EPS inner layer [24]. Summarizing with average thickness of the call wall of Gram-positive bacteria being ~50 nm [25,26], TB-EPSs are located at a distance of 50–1050 nm from the cytoplasmic membrane. LB-EPSs are more diffuse, form the outer EPS layer, are weakly bound with the inner layer and, apparently, are located at a distance > 290 nm from the membrane (thickness of the cell wall 50 nm + minimal thickness of the EPS inner layer 240 nm). Interestingly, those capsules, which also consist of extracellularly located polysaccharides, can reach a similar thickness, from 100 nm until 2500 nm, as shown, for example, for clinical isolates of *Streptococcus pneumoniae* [27].

The contents, compositions, and properties of EPSs produced by *Rhodococcus* cells were found to be different, as well as their influences on the flocculation, sedimentation, dewatering, and other capabilities [28,29]. The chemical composition is believed to afford some insights into the relationship between its chemical structure and function. The chemical nature of EPSs varies in terms of carbohydrates, proteins, nucleic acids, lipids, and humic substances (Figure 1). The dominant components of numerous *Rhodococcus* EPSs are carbohydrates and proteins (75–90%) [12]. The EPS also may contain carbohydrate and protein derivatives such as lipopolysaccharides, glycoproteins, and lipoproteins [30]. *Rhodococcus* actinomycetes synthesize different types of EPSs, such as acidic glucomanannes [31], heteroglycans [32], fatty acid-containing polysaccharides [33,34,35,36], or a complex of polymers [8,37]. The structure and completeness of EPSs reveal their three-dimensional structure, which becomes more stable and complex with increasing molecular mass because of multiple covalent bonds. EPSs of *Rhodococcus* spp. have various molecular weights ranging from several hundred thousand to several million daltons (Table 1). Below, specific components of EPSs produced by *Rhodococcus* actinomycetes are reviewed.

### 2.1. Carbohydrates (Exopolysaccharides)

The key structure-forming component of EPSs is carbohydrates, named ‘exopolysaccharides’. They are high-molecular compounds with a variety of carbohydrate compositions, consisting of monosaccharide residues forming linear or branched chains via diverse glycosidic bonds, which contributes a wide structural variety. It is noted that the type of links between both monomers and branching chains determine the physical properties of polysaccharides [38]. Variation in the composition of heteroglycans and their molecular weight may affect their ability to interact with proteins [39]. In the biofilm matrix, carbohydrates usually contribute to the stability and integrity and function as a protective barrier and pools for carbon and energy sources. However, most bacteria, unlike fungi, do not remetabolize their own polysaccharides [40]; this was confirmed for *Rhodococcus erythropolis* [41].

**Table 1 ijms-27-00498-t001:** The diversity of EPSs produced by *Rhodococcus* spp.

Species of *Rhodococcus*	Compound’s Name	Molecular Weight (Da)	Composition	Bioactivities	References
*R. hoagii* CECT555, *R. erythropolis* CECT3013, *R. rhodochrous* CECT5749, *R. rhodnii* CECT5750, *R. coprophilus* CECT5751	No data	No data	Contain histo-blood group antigens A (monofucosyl and difucosyl oligosaccharides), B (oligosaccharides with terminal galactose), and Lewis^y^ (difucosylated oligosaccharides)	Antiviral activity; binding norovirus virus-like particles	[42]
*R. pyridinivorans* ZZ47	No data	No data	No data	Antibiofilm, anti-angiogenic, antioxidant agents	[6,43]
*R. erythropolis* HX-2	HPS	1.04 × 10^6^	79.24% carbohydrates, 5.2% proteins and 8.45% lipids; Glucose, galactose, fucose, mannose and glucuronic acid with a mass ratio of 27.29%, 24.83%, 4.79%, 26.66%, and 15.84%	Anticancer and viscosity agents	[8]
*R. rhodochrous* ATCC 53968	No data	No data	Galactose, glucose, fucose, and glucuronic acid at a molar ratio of 3: 2: 2: 2 1.3% stearic acid, 4.1% palmitic acid, 5.8% pyruvic acid	Thickeners	[35]
*R. erythropolis* DSM 43215	PLS-1	1.14 × 10^6^	Glucose and mannose at a molar ratio of 1:1 3.3% of proteins	Anti-inflammatory agents	[41]
*Rhodococcus* strain 33	No data	1.05 × 10^5^	Glucuronic acid, glucose, galactose and rhamnose at a molecular ratio of 1:1:1:2.	Adhesion; improving biodegradation	[44]
*Rhodococcus* strain 33	33 EPS; PS-33	>2 × 10^6^	D-galactose, D-glucose, D-mannose, D-glucuronic, pyruvic acids at a ratio of 1:1:1:1:1	Improving hydrocarbon tolerance	[45]
*R. erythropolis* PR4	FR2	No data	d-galactose, d-glucose, d-mannose, pyruvic acid and d-glucuronic acid at a ratio of 1:1:1:1:1; 2.9% stearic acid and 4.3% palmitic acid	Improving hydrocarbon tolerance	[36]
*R. erythropolis* PR4	FACEPS	No data	Glucose, N-acetylglucosamine, glucuronic acid, and fucose at a molar ratio of 2:1:1:1.	Improving hydrocarbon tolerance	[33]
*R. rhodochrous* 202 DSM and *R. opacus* 89 UMCS	No data	No data	52.1% and 62.7% content of CH_x_ groups	Heavy metal (Ni(II), Pb(II), Co(II), Cd(II) and Cr(VI)) sorption	[46]
*R. opacus*	BES.DM-GMA-TETA-EPS Microspheres	No data	No data	Pb(II) and Cd(II) sorption	[47]
*R. erythropolis* ACCC 10543	RSF; NOC-1	No data	Proteoglycan (glycoprotein) composed of polysaccharides (91.2%), protein (7.6%), and DNA (1.2%).	Flocculation	[16,48]
*R. erythropolis* S-1 and 260-2	No data	No data	No data	Flocculation	[49,50]
*R. rhodochrous* R-202	R-202	1.3 × 10^6^	62.86% of polysaccharide and 10.36% of protein. Mannose, glucose, and galactose at a molar ratio of 12:6:1.	Flocculation (effective at pH around 7 and in salt solutions)	[51]
*R. opacus* 89 UMCS	No data	7.6 × 10^5^	64.6% polysaccharide 9.44% protein Mannose, glucose, and galactose.	Flocculation; Binding metal cations	[52]
*Rhodococcus* sp. *(erythropolis)* R3	No data	3.99 × 10^5^	84.6% protein and 15.2% sugar content	Flocculation	[53]
*R. qingshengii* QDR4-2	QEPS	9.450 × 10^5^	Mannose and glucose at a molar ratio of 81.5:18.5	Antioxidant agent; emulsification	[54]
*R. rhodochrous* ATCC 12674	SM-1 EPS	No data	D-galactose, D-glucose, L-fucose, and D-glucuronic acid at a molar ratio of 6:3:2:4 1.2% stearic acid, 2.3% palmitic acid, and 10.3% pyruvic acid	Absorption	[55]
*R. erythropolis* Au-1	No data	No data	No data	Emulsification	[56]
*Rhodococcus* sp. RHA1 (NRCC 6316)	No data	No data	Hexuronic acid and neutral glycose in the approximate ratio of 1:3. D-galactose, D-glucose, L-fucose, d-glucuronic acid at a ratio of 1:1:1:1	No data	[57]
*R. ruber* C208	No data	No data	Polysaccharides and proteins at a ratio of 2.5:1	Adhesion	[58]
*Rhodococcus* sp. SJ	EOM	No data	No data	Resuscitation of nonculturable cells; improving degradation of polychlorobiphenyls	[59]

In general, carbohydrates of EPSs are diverse. *Rhodococcus* spp. form heteropolysaccharides, mainly containing glucose, galactose, mannose, rhamnose, and sometimes uronic acids. For example, Urai et al. actively investigated acidic, fatty acid-containing polymers [33,35,36,55]. According to their results, exopolysaccharides from strains of *Rhodococcus rhodochrous* contain D-galactose, D-glucose, L-fucose, D-glucuronic acid, and sometimes mannose [34,35,55], and exopolysaccharides from *R. erythropolis* PR4 additionally contain D-mannose and pyruvic acid [36] or N-acetylglucosamine and fucose [33]. Polysaccharide PLS-1 from *R. erythropolis* DSM 43215 had glucose and mannose at a molar ratio of 1:1 [41]. Exopolysaccharide from *R. erythropolis* HX-2 (HPS) had 4.79% of fucose in its composition, in addition to glucose, galactose, mannose, and glucuronic acid, with a mass ratio of 27.29%, 24.83%, 26.66%, and 15.84%, respectively [8]. Mannose was shown to be the main residue in exopolysaccharides of *R. rhodochrous* R-202, *Rhodococcus opacus* 89 UMCS, and *Rhodococcus qingshengii* QDR4-2 [37,51,54]. It was exhibited that monosaccharide compositions with higher proportions of mannose are associated with better scavenging activity of the hydroxyl radical and consequently provide potential antioxidant capacity [54]. Exopolysaccharide PS-33 from *Rhodococcus* sp. and the other one from *Rhodococcus* strain 33 had another main monosaccharide and was composed of rhamnose, galactose, glucose, and glucuronic acid at a molar ratio of 2:1:1:1 [44,60]. Polysaccharide PS-33 is also acetylated and contained 6-deoxy sugars in the form of rhamnose, both in the backbone and in the side chain of the repeating unit. The presence of methyl groups and O-acetyl are likely play an important role in the emulsifying activity [60]. PS-33 was shown to consist of D-galactose, D-glucose, D-mannose, D-glucuronic, and pyruvic acids at a ratio of 1:1:1:1:1 [45]. Capsular polysaccharides of *Rhodococcus equi* serotypes were actively discovered by Richards’s research group. It was shown that they are mainly formed by glucose, mannose, glucuronic acid, rhamnose, and pyruvic acid [61,62,63,64]. EPSs of serotype 4 contain unique 5-amino-3,5-dideoxynonulosonic (rhodaminic) acid [63].

Most bacterial EPSs exhibit anionic characteristics due to the presence of uronic acids containing carboxyl groups, which are typical for most *Rhodococcus* strains [4] (Figure 1). Exopolysaccharides of *R. rhodochrous* R-202 consist of reducing sugars, uronic acids, and amino sugars at the concentrations of 232.41 μg/mg, 45.29 μg/mg, and 15.07 μg/mg, respectively [51]. The exopolysaccharide of *R. opacus* 89 UMCS had these components in similar proportions: reducing sugars, uronic acids, and amino sugars at concentrations of 184.79 µg/mg, 117.6 µg/mg, and 9.23 µg/mg, respectively [52]. The 33 EPSs were positioned as an acidic polysaccharide containing 66% neutral sugars, 18.4% uronic acids, and 15.6% pyruvic acid [45]. However, QEPS from *R. qingshengii* QDR4 did not have any uronic acids, according to FTIR spectroscopy and monosaccharide composition analysis [54]. For *Rhodococcus* sp. p52 grown in sodium acetate, it was revealed that viscous uronic acids accounted for the vast majority of the polysaccharides (72.08%) [65]. In our study, *Rhodococcus* strains of various species produced low amounts of total EPS carbohydrates, ranging from 0.6 ± 0.2 mg/L (*Rhodococcus aetherivorans* IEGM 1250) to 58.2 ± 24.0 mg/L (*Rhodococcus ruber* IEGM 231), with median EPS carbohydrate production equaling 8.9 mg/L [5].

In general, exopolymers are mainly acidic molecules because of the mixed contributions of polysaccharide- or protein-associated COOH−, NH^3+^, and phosphate groups. Exopolymers are treated as weak and surface-grafted polyelectrolytes, since long-chain macromolecules possess ionizable groups [24]. Dobrowolski et al. actually noted that high amounts of functional groups on the *Rhodococcus* extracellular polymers’ surface make interpretation of spectra more difficult [46]. Infrared spectrophotometry analysis revealed that the exopolymers contained carboxyl, hydroxyl, acetyl, and carboxylate groups, preferred for the flocculation and sorption processes [46,52]. In QEPS and HPS, a large amount of hydroxyl groups, and hydrogen bonding, causing the cross-linking of molecular chains, are detected [8,54]. The typical polysaccharide functional groups, namely, hydroxyl (–OH), amino (–NH_2_), and acylamino (–CONH_2_) groups, were all observed in the EPSs of *R. erythropolis* ACCC 10543 [16]. Amine and amide groups are also presented in *Rhodococcus* EPS [16]. According to the FTIR and XPS spectra analyses, exopolymer R-202 contains hydroxyl, carboxylic, amide, and amine groups, whereby fractions contain high amounts of oxygen and nitrogen, which may indicate a higher content of polar functional groups [51]. For EPS, from *R. rhodochrous* 202 DSM and *R. opacus* 89 UMCS, it was shown that 52.1% and 62.7% of carbon is present in –CH_x_ groups, respectively, and the presence of acetoamido or amino groups was indicated, in addition to main hydroxyl groups [46]. Obviously, the FTIR analysis of HPS showed strong peaks that characterized absorptions of C–O–C and C–O–H bonds of the pyranose ring and also peaks of methyl groups [8].

Aside from the components themselves, linkages between units also vary greatly. Pen et al. observed that the EPSs of rhodococcal cells at the stationary phase are regarded as amorphous random coils, probably due to the 1,2-α- or 1,6-α-linkages between their exopolysaccharides [24], and the EPSs of the late exponential phase, nevertheless, may possess 1,3-β- or 1,4-β-linkages between the exopolysaccharides, resulting in rigid helices that are less resilient to normal compression [66]. There are six key glycosidic linkages in QEPS: ← 2)-Manp-(1→, Manp-(1→, ← 2,6)-Manp-(1→, Glcp-(1→, ← 3)-Manp-(1→, and → 3) Glcp-(1→ [54]. What is suggested is that ← 3)-Manp-(1→ composed the main chain of QEPS and that ← 2,6)-Manp-(1→ indicated the existence of a 1→ 2,6 branching linkage. In PS-33, glucuronic acid, rhamnose, and glucose are linked at position 3, galactose at positions 3 and 4, and one rhamnose is a nonreducing terminus linked by (1→4) [60]. Extracellular polysaccharides of *Rhodococcus* sp. RHA1 is composed of tetrasaccharide repeating units linked in such a way that: d-GlcpA-(1→, → 3,4)-l-Fucp(1→, → 4)-d-Glcp-(1→, and → 3)-d-Galp-(1→) [57].

### 2.2. Proteins

EPSs contain several types of biopolymers, including carbohydrates, proteins, nucleic acids, and sometimes lipids. Depending upon the chemical composition of exopolymers, they might exhibit various capabilities, such as adsorption, biodegradability, and hydrophilicity/hydrophobicity, and thus find applications in various spheres. The main constituents of *Rhodococcus* EPSs are considered to be polysaccharides (up to 100%) and proteins (up to 85%); less frequently lipids (up to 89%, typically 1–10%); and in the least amounts nucleic acids, humic substances, and pyruvic acid (0.1–10%) (Figure 1, Table 1).

The proteins of exopolymers in the biofilm matrix contribute to its formation and stabilization and provide metabolism and mediate intercellular communication [7]. They are also supposed to support biofilm cohesion, adhesion, communication, and environmental response [7]. These proteins are usually extracellular enzymes, S-layer proteins, electron transfer (ferredoxin, rubredoxin) proteins, transmembrane transporters (porin), lectins, and stress response proteins [4,38]. In relation to practical properties, both proteins and polysaccharides play main roles in flocculation and biosorption. Their concentration and characteristics decide the fate of surface properties, as well as biodegradability of the EPSs [12].

The amounts of proteins vary in different species. For EPSs from rhodococci, it is known that the polysaccharides’ content is at least five times higher than the proteins’ content. For example, the total carbohydrate and protein contents in EPSs of *R. erythropolis* HX-2 were found to be 79.24% and 5.204%, respectively [8]. According to Czemierska et al., water-soluble fractions of exopolymers with flocculant activity contained six times less proteins than carbohydrates [51,52]. So, EPSs of *R. opacus* 89 UMCS have 64.6% polysaccharides and 9.44% proteins [52], and EPSs of *R. rhodochrous* R-202 have 62.86% and 10.36% of these components, respectively [51]. The extracellular polymer from *R. erythropolis* ACCC 10543 is described as a glycoprotein composed of 91.2% polysaccharides, 7.6% proteins, and 1.2% DNA [16,48]. In our work, concentrations of proteins in *Rhodococcus* EPSs were insignificant, with the maximum detected for *R. rhodochrous* IEGM 107 (1.131 ± 0.091 mg/L). Other strains were shown to produce less than 0.5 mg/L proteins [5]. A purified polysaccharide PLS-1 had a trace of glucosamine and 3.3% of proteins with predominant glutamic acid and glycine, followed by alanine, serine, and leucine in amino acid composition [41]. After purification, the lack of proteins and nucleic acids is detected in all ATCC 53968 EPSs, QEPSs, and SM-1 EPSs [35,54,55].

Protein content may be high or even predominant. In aerobic quinoline-degrading biofilms with a predominance of *Rhodococcus*, polysaccharides and proteins were determined as two primary components accounting for 78–87% of total EPSs [67]. Additionally, saccharides (57.0 ± 10%) and proteins (43.0 ± 6.9%) were the most dominant composition of TB-EPSs and LB-EPSs, respectively [67]. For bioflocculant from *Rhodococcus* sp. R3, it was revealed that the total protein and sugar content were 84.6% and 15.2% [53]. A significant proportion of proteins are also found in EPSs synthesized by *Rhodococcus* sp. p52 in the presence of dibenzofuran [65]. These compounds account for 20–33% of total EPSs and dominate polysaccharides by 1.5–3.0 times in LB-EPSs. The content of protein in the biofilm of *R. ruber* C208 was similar to that of polysaccharides, with a maximal content of 150–200 µg/mL [58]. Extracellular organic matter of a polychlorinated biphenyl (PCB) degrader *Rhodococcus* sp. SJ contains proteins and polysaccharides at concentrations of 24.33 mg/L and 1165 mg/L, respectively [59]. A protein-enriched fraction was optimized in the composition of *Rhodococcus* sp. SJ EPSs, since namely the proteins were shown to provide resuscitation-promoting activity [59]. Proteins of the optimized EPSs constituted 223.58 mg/L and had a molecular weight of more than 10 kDa. In its spectral analysis, tryptophan-like proteins, tyrosine-like proteins, and tyrosine-like amino acids peaks were observed. The presence of these proteins is likely related to microbial stress responses and stability maintenance, and polysaccharides may serve as supplementary nutrient sources that support microbial activity and contribute to degradation [59].

### 2.3. Lipids

Exopolymers of *Rhodococcus* spp. are supposed to possess hydrophobic properties due to polysaccharides-linked methyl and acetyl groups, lipids, and lipid derivatives in their composition [16]. Among exopolymers from *Rhodococcus,* a specific group of fatty acid-containing polysaccharides has been described [33,36]. According to Urai et al., saturated fatty acids such as stearic and palmitic acids prevail in the lipids of *Rhodococccus* EPSs [35,36,55]. It was demonstrated that fatty acids attach via ester bonds to the sugar backbone of exopolymers. So, FR2 EPSs from *R. erythropolis* PR4 contain 2.9% stearic and 4.4% palmitic acid [36]. Polymers from *R. rhodochrous* ATCC 53968 and *R. rhodochrous* ATCC 12674 (SM-1) have these fatty acids in proportions of 1.3% and 4.1% and 1.2% and 2.3%, respectively [35,55]. A mucoid strain of *R. rhodochrous* S-2 produces an extracellular polysaccharide of several million daltons in size, which contains stearic acid, palmitic acid, and oleic acid [34]. In HX-2 EPSs, from *R. erythropolis* HX-2, a total of 8.45% lipids were detected [8]. In our recent research, lipid-rich exopolymers of *Rhodococcus* IEGM strains were described [5]. It was shown that the presence of total lipids in EPSs is similar to, or prevails over, the carbohydrate content. In particular, production of these compounds varies from 15.6 ± 1.6 mg/L (*Rhodococcus* sp. IEGM 1401) to 71.7 ± 7.2 mg/L (*R. erythropolis* IEGM 1415), with a median value of 32.0 mg/L [5].

### 2.4. Humic Substances

Humic substances can comprise significant parts of dissolved extracellular organic matter [68] and can be divided into three components: fulvic acids, humic acids, and humin, which facilitate aggregation and cohesion [12]. According to the multiple research studies on *Rhodococcus* EPS composition, humic substances are rare or have not been defined. There is an example of detecting humic substances in exopolymers’ spectra. In extracellular organic matter of the *Rhodococcus* strain SJ, humic-like substances were identified [59]. However, it was shown that the optimization process substantially enhanced the protein content but reduced the relative abundance of humic-like substances. Tong et al. showed that deposition of microbial EPSs on silica surfaces could be significantly influenced by humic acids under solution conditions typical for subsurface environment, which could be applied in bioremediation [69].

## 3. Biological Functions

EPSs as major components of natural microbial systems play a substantial functional role in the physiology and environmental interactions of *Rhodococcus* cells. The physiological role of EPSs depends on the natural environment in which the microorganisms live. Since actinomycetes of the genus *Rhodococcus* live in a wide variety of habitats and are able to withstand extreme conditions [20], most of the functions ascribed to their EPSs are protective in nature. These biopolymers permit resistance under extreme environmental conditions, despite the fact that their synthesis and release require a lot of energy from the cells [24]. The basic role of polymers synthesized into surrounding media is to create and maintain favorable conditions for microorganisms. In this way, exopolymers form a hydrated barrier between the cells and the environment, reliably protecting the cells from adverse factors, firmly holding colonies to a solid surface, participating in the substrate assimilation, and contributing to the aggregation of microorganisms and formation of biofilms [70]. Rhodococci are known to be capable of producing biofilms that have been shown to be more resistant to hydrocarbons, metal nanoparticles, and other stresses. For example, the revealed adaptation mechanisms of *R. rhodochrous* IEGM 1363 include increased surface roughness and intensive formation of an exopolymer matrix rich in lipids, which plays the main role in protecting cells from reactive oxidative stress [71]. Our study confirmed protective functions in dense *R. ruber* IEGM 231 biofilms formed in the presence of toxic *n*-hexane and diesel fuel [5].

A special function of *Rhodococcus* exopolymers is considered to be the inactivation of chemical compounds, in addition to the effect on biofilm specificity [72]. Some authors have reported increased production of EPSs in the presence of toxic organic compounds. It was shown for *Rhodococcus pyridinivorans* XB during growth on di-(2-ethylhexyl)phthalate [73]. *Rhodococcus jostii* RHA1, in the presence of toluene and perfluorocarboxylic acid aqueous solution, produced exopolymers at a concentration above 2 mg/L [9]. EPS production was a typical adaptive reaction of this strain to various stresses [74]. *R. erythropolis* PR4, which degrades various alkanes, synthesizes a large amount of EPSs, which are likely provided the hydrocarbon tolerance of this strain [33]. Moreover, EPSs not only trap the toxic substances but also increase the adhesion of hydrophobic organic matter, thus improving the biodegradation efficiency [75]. An oxidation of anthracene or phenanthrene, for example, is associated with an attachment of rhodococcal cells to the substrate and the formation of biofilms on the surface of solid hydrocarbons [15].

Smooth colonies were supposed to be more resistant relative to rough ones due to produced EPSs. Research has shown that the presence of exopolymers altered the level of cell exposure to solvents. Iwabuchi et al. showed that some *R. rhodochrous* strains that were designated smooth were found to produce biofilms with mucoidal hydrophilic EPSs [76]. In contrast, rough strains of *R. rhodochrous* with a more hydrophobic surface were shown to be more sensitive. So, EPSs confer oil tolerance in members of *Rhodococcus* [76]. In the presence of organic solvents, part of the initially rough *R. erythropolis* DCL14 population started producing EPSs [77,78]. The DCL14 cultures were found to form cell aggregates and biofilms in the presence of both water-immiscible and water-miscible solvents [79,80]. Aizawa et al. also considered that the mucoidal morphology positively influenced the growth and survival of cells in the presence of xenobiotics. As a result, the addition of *Rhodococcus* sp. 33 EPSs to rough strains improved their tolerance to benzene [45].

It is considered that EPSs, as well, having certain architecture, provide an optimal environment for an exchange of genetic material between cells. Meanwhile, exopolysaccharides produced by *Rhodococcus* cells prevent binding of exogenous DNA by influencing horizontal gene transfer. Huang et al. confirmed that rich extracellular polymers prevent the entry of foreign DNA into cells, in the example of *R. ruber* YYL [81]. The YYL cells were shown to have mushroom-like substances surrounding them. So, the electrotransformation efficiency was significantly improved for mutant cells with the knocked out *gmhD* gene, responsible for the biosynthesis and assembly of capsular polysaccharides [81].

In fact, EPSs could also absolutely play other physiological roles. For example, EPSs of *R. equi* are determined as potential virulence factors, since they contribute to antigenic specificity and are recognized by the immune system [62,63]. Tripathi et al. highlighted a possible relationship between conjugation and biofilm formation, showing a mechanism by which the virulence plasmid can move among *R. equi* in the soil [82].

## 4. Applications (Bioactivities)

*Rhodococcus* cells’ unique biochemical properties and adaptability to harsh conditions make their metabolites valuable for bioremediation, industrial, and potential medical applications. EPSs are natural polymers that can be sustainably produced from renewable resources, and they possess a competitive advantage over other polymers [8]. In certain environments, bacteria can stably produce EPSs with high structural reproducibility; this is advantageous in terms of low fermentation costs, easy extraction, and short production cycles. Therefore, there is a huge commercial potential for EPSs in industrial applications. The properties of *Rhodococcus* EPSs in terms of their application areas are submitted further (Figure 2).

### 4.1. Biomedicine

Exopolysaccharides produced by *Rhodococcus* species are gaining attention for their promising biomedical applications due to their biocompatibility, non-toxicity, biodegrability, and also unique structural, physicochemical, and biological properties. Recent research highlights their potential in anticancer therapy, antioxidant activity, and bioemulsification and as biocompatible materials for pharmaceutical use.

*Rhodococcus* EPSs could have potential antiviral activity. For example, *Rhodococcus* spp. are able to attach to human norovirus particles using their EPSs. It was shown that exopolysaccharides derived from the five *Rhodococcus* strains bound to both GII.4 Sydney 2012 and GII.6 norovirus viral-like particles, but not to the rotavirus double-layer particles in the negative control [42]. According to recent studies, the exopolysaccharides of some *Rhodococcus* strains [8,33] consist of glucose, galactose, fucose, mannose, and glucuronic acid, which are sugar moieties also found in histo-blood group antigens in host cells and as such participate in molecular recognition. So, Santiso-Bellón et al. concluded that direct binding of norovirus to *Rhodococcus* cells appears via their exopolysaccharides, which led to the subsequent inability of these sequestered viruses to infect their target cells [42]. Blocking antibodies, which target A and B epitopes, reduced the binding of the EPSs from *R. erythropolis* to GII.6 VLP, while enhanced binding to GII.4 VLP was observed when A and B epitopes were blocked. Authors suggest that certain sugar residues do not participate in binding but promote the EPSs’ rigidity. While GII.6 VLP recognized the same epitopes on the EPSs as anti-blood group antibodies, GII.4 VLP recognized different epitope(s) whose flexible glycan structures were stabilized by the action of anti-HBGA antibodies, thereby increasing the affinity of GII.4 VLP. These investigations provide a basis for developing innovative antiviral strategies to prevent and treat NoV infections through EPSs [42].

Li et al. described QEPS extracted from *R. qingshengii* QDR4-2 with potential antioxidant activity [54]. The exopolysaccharide demonstrated the scavenging activities against 2,2-diphenylpicrylhydrazyl (DPPH), 2,2′-azino-bis(3-ethylbenzothiazoline-6-sulfonic acid), and hydroxyl and superoxide radicals that reached 30–49%. Moreover, QEPS could be used in pharmaceutical industries due to its good biocompatibility. Another exopolysaccharide from *R. pyridinivorans* ZZ47 demonstrated 54% and 22–27% DPPH and hydroxyl free radical removal, respectively [43]. QEPS has a high proportion of mannose and a smaller proportion of glucose (see Table 1). Hydroxyls are electron-donating groups of EPSs, which facilitate contact with free radicals [54]. Mannose residues in the form of α-D-mannopyranose contain two -OH groups next to each other (two and three carbon atoms) at one side of the molecule plane (https://www.kegg.jp/entry/C00936, last accessed 27 December 2025). As a hypothesis, this can provide a high density and regular distribution of these groups on the cell surface to catch many radicals. Uronic acids, however, are considered to provide better free radical scavenging activities of EPSs, stabilizing them and affecting the electron cloud density [54]. EPSs produced by *R. erythropolis* PR4, *R. rhodochrous* ATCC 53968, ATCC 12674, *Rhodococcus* sp. 33, and RHA1 harbor uronic acids (see Table 1) and can show promising antioxidant activities, which can be estimated in the future.

Exopolysaccharides could play a role in angiogenesis—a complex mechanism of the formation of new blood vessels, which is used for cancer treatment [83]. Güvensen et al. showed that exopolysaccharides from *R. pyridinivorans* ZZ47 have strong anti-angiogenic activity only at 2 mg/mL, so authors concluded that the exopolysaccharides have dose-dependent anti-angiogenic properties [6]. But, exopolysaccharides of *R. pyridinivorans* ZZ47 showed low cytotoxic activities on human colorectal adenocarcinoma (HT-29) and human breast adenocarcinoma (MCF-7) cells [43]. Moreover, *R. pyridinivorans* EPS showed antibiofilm activity between 11 and 55% against strains of *Salmonella typhimurium*, *Aeromonas hydrophila*, *E. coli*, *Pseudomonas aeruginosa*, *Shigella dysenteriae*, *Staphylococcus aureus*, and *Bacillus subtilis*. Due to the studied biological activities and lack of genotoxicity and cytotoxicity of the polymers, they may be suitable for use in pharmaceutical, diagnostic, and therapeutic industries [6].

Anti-angiogenic activity of the EPSs produced by *R. pyridinivorans* ZZ47 may be closely linked to their antioxidant activity. As mentioned in [6], microbial EPSs with antioxidant properties inhibited tumor angiogenesis-related genes’ expressions and upregulated anti-angiogenic genes in some cancer cell lines that can be related to the abruption of signal transduction pathways. Also, this can explain a dose-dependent mode of the ant-angiogenic effect of the ZZ47 EPSs, which is becoming stronger with proportional decreasing of free radical concentration. Antibiofilm activities of EPSs can be caused by degrading extracellular enzymes incorporated in the EPS matrix or their inhibiting effects on bacterial cell adhesion due to the presence of negatively charged functional groups (Figure 2). In our work [5], it was shown that the adhesion force of the cantilever of an atomic force microscope to crude EPSs produced by various *Rhodococcus* species was 1–3 nN lower than unmodified cover glass. Although no correlation was found between the EPS productions and adhesive activities of the studied *Rhodococcus* strains to polystyrene, these EPSs could condition solid surfaces and prevent adhesion of other bacteria to the conditioning film. Adhesion is the first step of the biofilm formation, and inhibition of this process results in stopping or reduced growth of microbial biofilms.

Bacterial exopolysaccharides are supposed to have a function of inhibiting cell proliferation [2]. HPS demonstrated a high inhibition effect on cancer cells without affecting the normal ones [8]. Hu et al. showed that adding HPS at a concentration of 800 μg/mL decreased the cell viability of A549, SMMC-7721, and Hela cells by 21.86%, 31.24%, and 37.65%, respectively, but HPS had no cytotoxicity effect on L929 cells independent of concentration [8]. Although, it was noticed that HPS can rapidly inhibit the proliferation of cancer cells on the first day and achieve a relatively high inhibition effect. On the one hand, *Rhodococcus* exopolysaccharides have been shown to exhibit anti-proliferative activity [8], while on the other hand, some exopolymers were non-cytotoxic to cancer cells [6]. We suppose that selective and dose-dependent cytotoxic activities of rhodococcal EPSs against tumor cell lines and their neutral effects on normal cells (fibroblasts L929 in work [8]) are related to specific ligand–lectin interactions. Tumor cells, for example, can express galactins (galactose specific receptors) and mannose receptors. Galactose and mannose are in the composition of HPS (Table 1). Moreover, HPS proteins can participate in specific EPS–cell interactions. These interactions can be a trigger for programmed cell death or inhibit cell proliferation. Summarizing revealed effects, *Rhodococcus* exopolysaccharides could be used for various therapeutic applications depending on the intended goals.

Some EPSs could provide biological activities such as anti-inflammatory effects [2]. This activity was confirmed for the extracellular heteropolysaccharide produced by *R. erythropolis* DSM 43215 [41]. These effects can have complex mechanisms and multiple causes, such as specific interaction of oligosaccharide motifs and proteins of EPSs with mannose-like, Toll-like, and other receptors in endothelial or immune cells; interaction with components of glycocalyx resulting in their binding (cationic inflammatory mediators, cytokines, complement, etc.) or changes in charge and viscosity; binding with lipopolysaccharide of pathogenic bacteria; chelating metal cations; and involvement in the reduction of oxidative stress, as discussed above. Although the heteropolysaccharide produced by *R. erythropolis* DSM 43215 did not contain a lipid fraction (see Table 1), lipids in EPSs produced by other *Rhodococcus* strains can potentially provide inflammatory activities through binding lipophilic prostaglandins.

In addition, potential biomedical applications of EPSs are associated with vaccine development. Rhichards et al. have undertaken a detailed chemical study of the capsular polysaccharide antigens of *R. equi* cells, suggesting that the development of protective vaccines against these pathogens are of considerable interest as a means of controlling disease [62]. Depending on the serotype, the capsular antigens contain β-D-mannose and α-D-galactose residues, in which the O-4 and O-6 positions are bridged by pyruvic acid acetal groups (serotypes 1 and 6) or carry lactic acid ether substituents at the O-3 position of α-L-rhamnose residues (serotype 2). Both of these acidic substituents occur in the serotype 3 polysaccharide, in which the pyruvic acid acetal groups are linked to O-2 and O-3 of β-D-glucose residues and lactic acid ethers are linked to O-4 of β-D-mannose residues. The serotype 7 polysaccharide is composed of the linear repeating trisaccharide units →3)-α-D-Gal-(l→3)-α-D-Man-(l→3)-α-L-Rha-(1→, in which the cyclic pyruvic acid acetal groups bridge the O-4 and O-6 positions of the α-D-mannose residues showing the S-configuration. A repeating unit of the serotype 4 capsular polysaccharide is →3)-β-D-Man-(l→4)-β-D-Glc-(l→4)-α-D-Glc-(l→4)-α-RhoANAc-(2→, where RHoANAc is unique rhodaminic acid, in which the cyclic pyruvic acid acetal groups bridge the O-7 and O-9 positions. In addition, virulence proteins VapABCDE are antigens, which are well recognized by the immune system [62,63,64]. Diversity of antigens, most probably, requires the development of polyvalent (multicomponent) vaccines.

Exopolymers synthesized by rhodococcal cells have the potential to act as gelling agents, thickeners, emulsifiers, stabilizers, and water binders [35,54]. Hu et al. described a unique exopolysaccharide from *R. erythropolis* HX-2, which exhibited a stereoscopic network structure [54]. Due to its small pore size (<1 μm), dense distribution, and aggregation of polysaccharides, the HPS may retain more water molecules, which makes them good excipients. It was shown that the water solubility index and water-holding capacity of HPS were 92.15 ± 3.05% and 189.45 ± 5.65%, respectively. So, HPS may have excellent hydrophilicity and the potential to hold a mass of water through hydrogen bonding. The solution viscosity of HPS equaled 18 mPa·s in 1 mg/mL and 30 rpm, which was lower than the viscosity of xanthan gum (61 mPa·s). Another exopolysaccharide from *R. rhodochrous* ATCC 53968 features high viscosity, with a value amount of 1.39 m^3^/kg [35]. Medium and high viscosity indicates a thickening capacity of *Rhodococcus* exopolysaccharides, which was more beneficial for the application of excipients to the health industry. There is another exopolysaccharide with excellent moisture retention and absorption capacities that was described by Urai et al. [55]. SM-1 EPS from *R. rhodochrous* ATCC 12674 was capable of retaining the initially added water and also absorbing moisture in the desiccator under dry and high-temperature conditions. The authors claimed that SM-1 EPSs’ absorption capability was much higher than that of the other moisture absorbents tested, such as silica gel and hyaluronic acid.

Likely, lipid-rich EPSs produced by *Rhodococcus* cells (see Table 1) are not really promising to retain water and be applied as thickeners, as lipids make EPSs more hydrophobic and water-repellent. However, lipids can provide the emollient and skin softening properties of EPSs. In a couple with the presence of low molecular organic acids, namely pyruvate used for chemical peeling, EPSs from *Rhodococcus* cells can be promising for cosmetic medicine.

It is important to note that most studies of biological activities of EPSs produced by *Rhodococcus* bacteria were performed in vitro. Only in one work were the anti-angiogenic effects of EPSs produced by *R. pyridinivorans* ZZ47 estimated on embryos in fertilized chicken eggs and a single-dose acute toxicity test conducted using laboratory mice [6]. According to this single study, EPSs of *Rhodococcus* are not toxic and can be used to prevent a blood supply of tumors. On the other hand, a revealed dose-dependent anti-angiogenic activity is a concern for use of ZZ47 EPSs in pregnant women (because of possible adverse effects on embryos) or at regeneration of tissues, and their local application can be a better option. However, there is a lack of sufficient information to declare therapeutic effects and biosafety of EPSs produced by *Rhodococcus* cells, which can be a direction for future investigations. A deep purification of EPSs with controlled chemical composition is required to apply these compounds in medicine, which must be taken into account in the development of EPS-based pharmaceuticals.

### 4.2. Bioremediation

*Rhodococcus* actinomycetes are known biodegraders, capable of decomposing xenobiotics to inorganic products or low molecular organic fragments. Their EPSs could be actively used in bioremediation, since they enhance adhesion and biodegradation, have emulsification activity, and, moreover, protect cells against pollutant adverse effects. EPSs, as cell-secreted polymers, influence the biofilm formation and the surface characteristics of bacteria, in turn positively affecting adhesion between the bacteria and contaminants, which leads to improving degradation efficiency. Notably, the negatively charged functional groups presented in EPSs have electrostatic interactions with various ions and pollutants in the surroundings and help in their biosorption [84].

EPSs encapsulating bacterial cells adhere to and dissolve hydrophobic organic matter, thereby promoting the utilization of the hydrophobic substances [85]. For example, EPSs secreted by *Rhodococcus* sp. p52 affected both the biofilm formation and the dibenzofuran degradation [65]. It was shown that the absolute value of the zeta potential of p52 cells decreased with the degradation and the production of EPSs, and this was beneficial for increasing the adhesion of strain p52 to dibenzofuran. Both tight and loosely bound EPSs produced by strain p52 increased the solubility of the pollutant in water and thus promoted the degradation. The authors assumed that the presence of dibenzofuran increased the cellular stress of strain p52, which in turn led to the secretion of more extracellular polysaccharides. Not only was the protective function of EPS important for dibenzofuran degradation, the amount of electron transfer proteins (rubredoxin and ferredoxin) increased in the presence of this pollutant. It is known that ferredoxin can transfer electrons to dioxygenases, which contributes to the subsequent degradation of dibenzofuran [65].

Shi et al. discovered that *Rhodococcus* strain SJ, isolated from PCB-contaminated soil, enters the viable but nonculturable state under stress, and there is the possibility of its secreted exopolymers to facilitate resuscitation from this state and enhance degradation [59]. PCB degradation efficiencies in EPS-amended cultures reached 90.0%, 84.5%, and 80.7% at PCB concentrations of 1, 5, and 10 mg/L, respectively, compared with 80.5%, 65.9%, and 46.9% in the control, unamended cultures. It was indicated that the EPS protein fraction significantly enhanced the PCB degradation ability of strain SJ, with a stronger effect observed at higher PCB concentrations. Furthermore, it was concluded that utilizing EPSs in stress recovery strategies for sustainable bioremediation represents a more practical and cost-effective approach compared with the use of purified resuscitation-promoting factors, a class of quorum-sensing signaling molecules.

Semeniuk et al. described the exopolysaccharide from *R. erythropolis* Au-1 with potential as an effective emulsifying agent [56]. The emulsifying activity of its solutions with vaseline oil has been found to equal 42–58%, depending on the exopolymer’s concentration. The SM-1 EPS forms a water-in-oil emulsion and prevents evaporation of water under dry, high-temperature conditions, since SM-1 EPS is an efficient emulsifier of oil, polyaromatic hydrocarbons, and alkanes [55]. There is another example of EPS emulsification activity. The S-2 EPS certainly enhanced the abilities of native marine bacteria to degrade some components of aromatic oil fractions [34]. It was shown that, when S-2 EPS was added to the medium, oil was immediately emulsified, whereas it either adhered to the inner surface of the culture tubes or formed oil clumps in the liquid when S-2 EPS was not added. The authors concluded that EPS produced by *R. rhodochrous* S-2 could be useful for the biodegradation of spilled oil in marine environments, and especially for the bioremediation of polyaromatic hydrocarbons, that remain in the environment even after a traditional bioremediation treatment [34]. In another research, it was noted that S-2 EPS protects rough *Rhodococcus* strains from the toxic oil [76].

EPS synthesis was proposed to be a crucial mechanism for the assimilation of alkane substrates at low temperature by *Rhodococcus* sp. strain Q15, and it was described to have an emulsifying effect on alkane assimilation and oxidation [86]. With close contact between the strain Q15 cells and hydrocarbon surfaces achieved, extracellular compounds could then solubilize the alkane substrates, facilitating cellular uptake. The EPS may also play a role in floc formation in strain Q15, enabling the cells to remain in close physical contact with hydrocarbons. Such phenomenon was described by Takeda et al. for biopolymer produced by *R. erythropolis* S-1 during growth in pentadecane [87].

The biodegradation process starts with attachment of bacteria to the substrate. To facilitate adhesion to hydrophobic substrates, hydrocarbon-degrading bacteria may increase cell surface hydrophobicity by modifying cell surface components [88]. EPSs are supposed to be one of the major causes of *Rhodococcus* cell surface hydrophobicity [44,60]. Urai et al. measured *R. rhodochrous* cell hydrophobicity, obtaining overestimated values for cells surrounded by loosely bound material that could be easily removed by physical impact, such as by stirring [55]. In addition, microbial cells may produce EPSs in the form of capsules or mucoid secretions that may interact with hydrophobic substrates, such as hydrocarbons [89]. Polysaccharides carrying hydrophobic groups are involved in the labelling of hydrophobic interfaces by microorganisms [60]. It is known that polysaccharides rich in deoxy sugars engage in the adhesion of bacteria to hydrophobic interfaces [90]. Neu et al., in another research, described emulsion-stabilizing exopolysaccharide from the adhesive, hydrophobic *Rhodococcus* strain No. 33, which consists of different deoxy sugars [60]. Neu and Poralla extracted the amphiphilic polysaccharide from *Rhodococcus* sp. and confirmed that polymers contribute cell surface hydrophobicity and play a role in adhesion [44]. Meanwhile, S-2 EPS has been shown to lower the cell surface hydrophobicity of rough strains, indicating that S-2 EPS functions as a hydrophilin and thus establishes tolerance to oils and *n*-hexadecane [91].

Lipids seem to play an important role in hydrophobicity and emulsifying activities of EPSs produced by *Rhodooccus* bacteria. Although, few studies measured the amounts of these compounds in EPSs. A total of 7.2% fatty acids was found in LB-EPSs produced by *R. erythropolis* PR4, the strain tolerating hydrocarbons [36]. Most studies have a focus on the polysaccharide composition of EPSs (see Table 1), and physical-chemical methods (cell adhesion to hydrocarbons and zeta-potential measurements) are commonly used to estimate hydrophobicity without biochemical analysis. We assume that not only quantities of lipids can be important to provide an optimal hydrophilic–lipophilic balance of EPSs, but also their equal distribution in the EPS matrix and exposure to hydrophobic molecules, i.e., number and density of hydrophobic sites on the surface of EPSs or EPS-surrounded rhodococcal cells.

EPSs of actinomycetes are also known as sorption agents for heavy metals. Exopolymers can act as adsorbents because of their particular structure, physicochemical properties, and chemical stability, which are the result of the presence of chemical reactive groups in polymer chains. In the case of heavy metals, the hydrophilicity is an essential property [92]. Such a characteristic of EPSs is attributed to chemical composition and culture conditions. Dobrowolski et al. described EPSs of *R. rhodochrous* and *R. opacus*, which exhibited high adsorption affinity towards Cd(II), Pb(II), Ni(II), Co(II), and Cr(VI) ions, with the highest adsorption capacities obtained for Pb(II) and Cd(II) ions [46]. A rapid rate of ion adsorption is found to be a significant advantage of EPSs, which allows them to be used in flow systems used for wastewater treatment [46]. In another study, Dobrowolski et al. used immobilized EPSs for adsorption of Pb(II) and Cd(II) from aqueous solutions [47]. EPSs extracted from *R. opacus* were immobilized on synthetic microspheres BES.DM-GMA-TETA, obtained via copolymerization of bis[4(2-hydroxy-3-methacryloyloxypropoxy) phenyl]sulfide with glycidyl methacrylate and modified with triethylenetetramine. Such a fixing method is supposed to improve the solid phase properties and sorption capacities of the crude EPS. In the result, the coverage of the EPS on the synthetic microspheres was only 6.25%, but it enabled increase the adsorption capacities for both Pb(II) and Cd(II) by 47% and 25%, respectively [47]. It is important to note that not only extracted and immobilized EPSs can be used for removal of heavy metals, but growing *Rhodococcus* cells are promising as well. Rhodococci are highly tolerant to heavy metal salts with minimal inhibitory concentrations ≥ 200 mg/L [93,94,95] and can be used for ion accumulation via the binding, with EPS at concentrations used in works [46,47] being 5–700 mg/L.

EPSs have a good flocculation activity [13]. Bioflocculants are usually effective in aggregating colloids and are widely used not only in remediation fields, but in industrial fields such as tap water preparation, downstream techniques, fermentation process, and food industries [96]. *Rhodococcus* strains have been described as effective flocculant producers. For example, various exocellular polymers are produced by *R. erythropolis* ACCC 10543 in different conditions. Peng et al. cultivated *R. erythropolis* on sludge and livestock wastewater and obtained bioflocculant RCF, which was effective over a wide pH range from 2 to 12, with flocculating rates being higher than 98% [16]. Another bioflocculant, NOC-1, is produced by *R. erythropolis* ACCC 10543 in standard conditions, and unlike RCF, has only 30% flocculating rates at pH 2.0 [16,48]. Kurane et al. described bioflocculants from *R. erythropolis* S-1 and *R. erythropolis* 260-2 and reported that *Rhodococcus* flocculants have very interesting, unique flocculation characteristics, but low productivity [49]. In addition, exopolymers R-202, synthesized by *R. rhodochrous* with the highest flocculating activity at the pH value 7.2 and 10 mM concentration of salt solutions, were obtained by Czemierscka et al. [51].

For heavy metal sorption and flocculation, not only is the number of functional groups important, but also their density and availability. Compact and dense EPSs have a smaller available contact surface, while diffuse EPSs with extended exopolysaccharide brushes or chains can interact with many ions or colloid particles [24]. Negatively charged uronic acids harboring -COOH groups are key binders of metal cations, while proteins are most important for flocculation (Table 1, Figure 2). EPSs with flocculating activities are composed of 7% to 85% proteins (Table 1).

*R. opacus* 89 UMCS cells with the affinity for calcite and magnesite surfaces produce exopolymers able to bind kaolin particles, resulting in flocculation [52]. It was noted that the soluble fraction of EPS had a molecular weight of 760 kDa. This parameter is important for flocculation, so this process with a high-molecular weight bioflocculant involves more adsorption points, stronger bridging ability, and higher flocculating activity [97]. Also, this macromolecule can interact with positively charged functional groups of particles suspended in water, causing their aggregation, as well as binding metal cations, e.g., Ca^2+^, Fe^2+^, and Mg^2+^. The above confirms that bioflocculants from *Rhodococcus* EPSs are suitable for the mechanical treatment of water, for example, water with soil or clay particles.

At bioremediation, especially in open and fragile ecosystems, evaluation of ecological risks related with bioremediation agents is essential. Although all EPSs are mixtures of biopolymers, which can be degraded by microorganisms, their fate in the environment, the time of life, and impacts on ecosystems can result in significant effects. As efficient sorbents and traps for metal ions and organic molecules, they can adsorb and retain dangerous toxicants acting as a sink and secondary contamination. This was shown for lake-derived EPSs of sludge, which tightly bound perfluorooctanoic acid and perfluorooctane sulfonate. Stable self-assembled complexes were formed, altering EPS conformation and protein secondary structure. The binding with N-containing compounds (EPS proteins and amino sugars) resulted in the formation of compact multilayered shells and the persistence of these fluoroalkyl substances. Phytotoxicity tests indicated great environmental risks of these complexes and their inhibiting effects on plants [98]. The EPSs used separately from producing *Rhodococcus* biodegraders can prevent biodegradation of contaminants, binding and holding their molecules. It was shown for nonylphenol, which resulted in the formation of aromatic, humified, and hydrophobic tyrosine-like, tryptophan-like, and soluble microbial by-product-like substances in colloidal EPSs. These complexes were toxic for sedimental bacteria, and degradation of nonylphenol was reduced. The addition of Fe_2_O_3_ nanoparticles softened these negative effects [99]. On the other hand, binding of toxicants reduces their migration and acute toxicity to living organisms. It should be taken into account that application of EPSs alone or bound with live *Rhodococcus* cells for sorption and chelating heavy metals does not provide elimination of these emergent pollutants. After degradation of EPSs and the death of *Rhodococcus* producers, heavy metal ions release into the environment again. The development of technologies where *Rhodococcus* EPSs with accumulated heavy metals can be replaced and processed further ex situ can help to overcome this inconvenience.

Another concern is the possible antimicrobial activities of *Rhodococcus* EPSs applied in large amounts, which can affect the indigenous microorganism. In the literature, these activities of microbial EPSs are considered to be beneficial and prevent food spoilage and biofouling [100,101]. In our work [5], the presence of a big proportion of lipids in LB-EPSs produced by *Rhodococcus* spp. has been found. These EPSs can supply microorganisms with additional available nutrients, promote the growth of satellite species, and stimulate the formation of a developed microbial food web. It is recommended to estimate complex ecological effects of *Rhodococcus* EPSs before their application. The total ecological risk will also depend on the form of the EPSs (soluble or cell-bound) and the system (open or contained) where they are applied. Additionally, pathogenicity of strains must be addressed in case viable *Rhodococcus* producers are introduced. Species of the genus *Rhodococcus* belong to the risk group 1 (https://lpsn.dsmz.de/genus/rhodococcus, last accessed 24 December 2025), excluding one opportunistic species *R. equi* (https://lpsn.dsmz.de/species/rhodococcus-equi, last accessed 24 December 2025) and a phytopathogenic species *R. fascians* [102].

### 4.3. Other Applications

Since the microbial EPSs possess different sugar compositions and different rheological characteristics, they represent a very big potential reserve within which it is possible to find the ideal EPS to use in a specific biotechnological and bioengineering application [11,103].

*Rhodococcus* bacteria could produce high-molecular exopolymers with potential in the food industry. The high molecular weight of the QEPS reached 9.450 × 10^5^ Da, which may contribute to its excellent emulsifying capabilities [54]. The emulsification indexes of QEPS for all six edible oils at a concentration of 2 mg/mL were between 54 and 73%, with a higher index for olive oil. Moreover, QEPS is suggested to have potential as a bioemulsifier and antioxidant with applications in the health, food, and pharmaceutical industries. The HPS exhibited higher molecular weight, which is equal to 1.04 × 10^6^ Da, which suggests that HPS may have a higher viscosity in addition to high water solubility [8]. Therefore, these hydrophilic exopolysaccharides may retain more moisture, which is a perfect characteristic for excipients in the food industry. Exopolysaccharide PLS-1 from *R. erythropolis* DSM 43215 also had an unusually high molecular weight, amounting to 1.14 × 10^6^ Da [41]. The total exopolymer and the fraction of the water-soluble exopolymer from *R. opacus* show a fibrillar structure with a sheet-like texture. This indicates the thin web structure of these preparations with higher capillary forces to hold water molecules [52]. Additionally, these polymers have flocculation activity, which can be applied in food and fermentation industries for removing pollutants.

In terms of bacterial crystallization processes, the production of specific bacterial outer structures and their chemical nature might be crucial factors, influencing the mineralogy and morphology of calcium carbonate crystals [104]. An important aspect in biomineralization processes is the role played by functional groups at the surface of the organic matrix, produced by microbial cells, especially acidic carboxyl and phosphate groups, which may bind cations such as Ca^2+^ and hence act as the crystal nucleation sites [105]. It was found that the EPSs from *R. opacus* strains are capable of binding calcium ions and hence act as nucleation centers influencing calcium carbonate precipitation, but the exopolymers do not affect the crystal structure [37,105]. Bacterially produced carbonate biominerals are known to be used for improving the durability of buildings, remediation of water or soil environments, and sequestration of atmospheric CO_2_ [103].

## 5. Influence of Growth Conditions and Extraction Methods on Composition and Properties of *Rhodococcus* EPSs

Major concerns of widely use EPSs produced by *Rhodococcus* bacteria are variations in their chemical composition and, therefore, in their activities and properties, as well as non-universal methods of extraction and purification. These procedures are essential steps for investigating the structure and biological activities of EPSs. Table 2 shows the diversity of methods used to extract and purify EPSs in the examples considered. Together, these studies demonstrate that both cultivation conditions and downstream processing strongly influence the yield (Table 2), composition, and properties of *Rhodococcus* EPSs (Table 1), which in turn complicates comparison between studies and limits industrial scalability.

**Table 2 ijms-27-00498-t002:** Growth conditions, extraction and purification methods, and their influence on the EPS production by *Rhodococcus* spp.

*Rhodococcus* Strains	EPS Name	Growth Conditions	Extraction Method	Purification Method	Yield of EPS	Reference
*R. hoagii* CECT555, *R. erythropolis* CECT3013, *R. rhodochrous* CECT5749, *R. rhodnii* CECT5750, *R. coprophilus* CECT5751	No data	TSA or YEME medium, 26/30 °C, 200 rpm, overnight	Centrifugation 12,000× *g*; addition of 3 volumes of anhydrous ethanol; centrifugation 12,000× *g*, 20 min, 4 °C	Precipitation of proteins with TCA	No data	[42]
*R. pyridinivorans* ZZ47	No data	TSB, 37 °C, 200 rpm, 24 h	Centrifugation, 10,000× *g*, 15 min, 4 °C; precipitation with 2 volumes of ethanol, 4 °C, 24 h; centrifugation 10,000× *g*, 20 min, 4 °C	Precipitation of proteins with TCA	10.136 g/L	[6,43]
*R. erythropolis* HX-2	HPS	Minimal salt medium (Na_2_HPO_4_ 1.5, KH_2_PO_4_ 3.48, (NH_4_)_2_SO_4_ 4.0, MgSO_4_ 0.7, yeast powder 0.01, 2% (*v*/*v*) sodium citrate culture), 25 °C, 72 h.	Heat, 90 °C, 15 min; centrifugation 10,000× *g*, 15 min, 4 °C; addition of 70% (*w*/*v*) TCA with a final concentration 15% (*w*/*v*), 4 °C, overnight; centrifugation 11,000× *g*, 20 min, 4 °C; precipitation with 3 volumes of 95% ethanol, 4 °C, 12 h; centrifugation 11,000× *g*, 20 min, 4 °C	Anion exchange chromatography on a DEAE-Cellulose DE-52 column; dialysis; gel filtration on a Sepharose CL-6B column	6.365 g/L (crude EPS)	[8]
*R. rhodochrous* ATCC 53968	No data	IB agar, 37 °C.	Shaking, 120 rpm, 10 min, 25 °C; centrifugation 10,000× *g*, 10 min; ethanol precipitation	Phenol/chloroform treatment; treatment with DNase I, RNase A, proteinase K, 37 °C, 16 h; dialysis	0.017 g/g of cells	[35]
*R. rhodochrous* ATCC 12674	SM-1 EPS				No data	[55]
*R. erythropolis* DSM 43215	PLS-1	Basal medium B with 1% (*w*/*v*) of a carbon source (*n*-alkanes, lower mono-, di- and trihydric alcohols, sugars), 30 °C, 24 h	Centrifugation; precipitation with 2 volumes of ethanol; washing with a small amount of 30% ethanol; dissolving in water; precipitation is repeated twice	Precipitation with 10% aqueous solution of N-cetyl-N, N, N-trimethylammonium bromide; washing with water and dissolving it in 5 M NaCl; dialysis, room temperature, 2 days; gel Filtration by Sepharose 4 B and 6 B columns	0.003–1.935 g/L	[41]
*Rhodococcus* strain 33	No data	Davis–Mingioti medium (1% *n*-hexadecane, yeast extract 0.1, K_2_HPO_4_ × 3H_2_O 4.8, KH_2_PO_4_ 1.5, (NH_4_)_2_SO_4_ 1.0, trisodium citrate × 2H_2_O 0.5, MgSO_4_ × 7H_2_O 0.2, trace elements), 20 °C.	No data	No data	No data	[44]
*R. erythropolis* PR4	FR2, FACEPS	IB agar, 25 °C.	Centrifugation	CTAB method; ethanol precipitation; DEAE-Toyopearl 650 M column chromatography	No data	[33,36]
*R. rhodochrous* 202 DSM and *R. opacus* 89 UMCS	No data	Liquid medium (glucose 20, KH_2_PO_4_ 2.0, K_2_HPO_4_ 5.0, NH_4_Cl 0.5, NaCl 0.1, MgSO_4_ 0.1, yeast extract 0.5), 26 °C, 130 rpm, 72 h	Centrifugation twice; concentration by a reverse osmosis process; centrifugation; filtration using a Durapore membrane (0.45 μm diameter of pore, Millipore, Burlington, USA); adding 95% ethanol, 72 h, 4 °C; centrifugation; precipitation is repeated twice	Dialysis, 4 °C, 3 days	No data	[46]
*R. opacus*	BES.DM-GMA-TETA-EPS Microspheres				No data	[47]
*R. opacus* 89 UMCS	No data	Liquid medium, 26 °C, 130 rpm, 10 days.			0.266 ± 0.046 g/L	[52]
*R. rhodochrous* R-202	R-202	Liquid medium, 28 °C, 130 rpm, 5 days.			0.273 ± 0.025 g/L	[51]
*R. erythropolis* ACCC 10543	RSF; NOC-1	Standard medium (sucrose 20.0; urea 4.0; yeast powder 1.0, NaCl 1.0, K_2_HPO_4_ 5.0, KH_2_PO_4_ 2.0, MgSO_4_ 0.2), 30 °C, 130 rpm, 20 h	Centrifugation 6000 rpm, 4 °C, 10 min; adding two volumes of acetone or cold ethanol; centrifugation	Washing with ether	1.600 g/L	[16]
*R. qingshengii* QDR4-2	QEPS	MRS broth, 25 °C, 150 rpm, 72 h	Heat, 90 °C, 30 min; centrifugation 11,000× *g*, 4 °C, 10 min; concentration using a rotary evaporator (RE-52AA, Shyarong), 45 °C, −0.1 MPa; precipitation with 3 volumes of ethanol, 4 °C, overnight; centrifugation 11,000× *g*, 4 °C, 15 min.	Precipitation of proteins by adding 75% TCA (*w*/*v*) to a final concentration of 15%; centrifugation, 4 °C, 11,000× *g*, 30 min; DEAE cellulose-52 column chromatography	3.850 g/L	[54]
*R. erythropolis* Au-1	No data	Liquid nutrient medium (NaNO_3_ 3.0, K_2_HPO_4_ 2.0, KH_2_PO_4_ 2.0, MgSO_4_ × 7H_2_O 0.5, Na_3_C_6_H_5_O_7_ × 2H_2_O 1.0, yeast extract 1.0, glycerol 20.0), 30 °C, 220 rpm, 5 days	Precipitation with 2 volumes of 96% ethanol	Reprecipitation from distilled water twice	5.0 g/L	[56]
*Rhodococcus* sp. RHA1 (NRCC 6316)	No data	Brain heart infusion, 30 °C, 175 rpm	Centrifugation, 60 °C; addition of 90% aqueous phenol (60 °C); centrifugation, 5 min	Dialysis, 4 °C; treatment with RNase, DNase, proteinase K (37 °C, 4 h); low speed centrifugation; ultracentrifugation (105,000× *g*, 4 °C, 12 h)	3.5% based on dry cell weight	[57]
*R. ruber* C208	No data	Synthetic medium (NH_4_NO_3_ 1.0, K_2_HPO_4_ 1.0, MgSO_4_·7H_2_O 0.2, KCl 0.15, CaCl_2_·2H_2_O 0.1, microelements), 30 °C, 150 rpm	Incubation of removed film with 2% (*v*/*v*) aqueous SDS solution, 4 h; centrifugation 34,800× *g*, 10 min	Dialysis	No data	[58]
*Rhodococcus* sp. SJ	EOM	Lactate minimal medium, 30 °C, 160 rpm, 216 h	Centrifugation 8000 rpm, 15 min	Filtration, 0.22 μm membrane	No data	[59]

Compositions of media are represented in g/L. TCA—trichloroacetic acid.

In the studies reviewed, the average *Rhodococcus* strains are cultivated in minimal or mineral media with various carbon and energy sources added (i.e., *n*-alkanes, such as *n*-hexadecane; alcohols, such as glycerol, ethanol, methanol, ethylene glycol, *n*-propyl alcohol, *sec*-propyl alcohol, 1,2-propandiol, 1,3-propandiol, *n*-butyl alcohol, *sec*-butyl alcohol, *tert*-butyl alcohol, 1,3-butandiol, 1,4-butandiol; and sugars, such as glucose, sucrose and fructose) (Table 2). In some experiments, rich nutrient media were used [6,33,42,54,57] (Table 2). Many studies demonstrate that EPS synthesis in bacteria, including actinomycetes, is highly responsive to the nature of the carbon source and other cultivation parameters [3,106,107]. *Rhodococcus* EPSs are arguably produced on sugars differing to some extent from those synthesized during growth on alkanes, alcohols, or complex substrates. This metabolic flexibility, while advantageous from a physiological perspective, results in EPSs with heterogeneous monosaccharide composition, molecular weight, degree of branching, and presence of non-carbohydrate moieties, such as fatty acids or proteins. Specific examples can be found below.

For *Rhodococcus* sp. p52, it was shown that the composition and proportion of extracellular polysaccharides greatly varied with the carbon source (dibenzofuran or sodium acetate) [65]. In the case of using dibenzofuran as a carbon source, glucose (74.25%) was prevalent in monosaccharide composition. But, when using sodium acetate, exopolysaccharides consisted of guluronic acid (25.34%), mannuronic acid (24.09%), and galacturonic acid (22.65%) in approximately equal proportions [65]. It was defined further that the proportion of antitoxin proteins in EPSs produced by *Rhodococcus* sp. p52 was five times greater in toxic conditions (namely, in the presence of dibenzofuran) than when using sodium acetate as a carbon source. The percentages of cold shock proteins and DUF4193 domain-containing proteins, which are associated with the stress response, also greatly increased [65]. The EPSs produced by *Rhodococcus* sp. 33 differ in research studies [45,60], which is evidence for the fact that a conformation of EPS chains is easily affected by conditions of the surrounding medium [24].

The component structure of the extracellular matrix of *Rhodococcus* biofilms was also changeable in different conditions, thus being a kind of adaptation. The increased content of polysaccharides in the matrix was observed in the periphery of the cell conglomerates of the *R. rhodochrous* IEGM 1363 biofilms at the maximum CuO nanoparticles concentration, while lipid content remained stable when biofilms were exposed to increasing nanometal concentrations. Similarly, increased polysaccharide content was detected in *R. ruber* C208 biofilms grown on a polyethylene surface under nutrient-limited conditions [58]; meanwhile, an increased protein content of this strain’s biofilms was revealed when growing on polystyrene [108]. Moreover, the content of proteins in the extracellular matrix of *R. ruber* C208 biofilms after 20 days of incubation was up to 2.5-fold lower than after 10 days [58]. That was evidence for the influence of the cell culture age on the composition of EPSs produced by *Rhodococcus* bacteria.

On the other hand, for example, the authors confirmed that the composition of EPS PLS-1, unlike the production, is independent of the nature of their carbon and energy sources [41], agreeing with Sutherland’s statement about the composition of many extracellular heteropolysaccharides [109]. In Rapp et al.’s research, the influence of cultivation conditions on PLS-1 exopolysaccharide production by *R. erythropolis* DSM 43215 was shown [41]. Of all the lower mono-, di-, and trihydric alcohols, sugars, and *n*-alkanes used, glycerol was discovered to be the best substrate for polysaccharide formation. In that variation, the highest yield of PLS-1 amounted to 947 mg/L at 1.25% (*w*/*v*) concentration. Meanwhile, diammonium hydrogen phosphate was most effective among inorganic nitrogen sources used, and yeast extract among natural nutrients. The amounts of exopolysaccharides in these cases were 750 and 245–601 mg/L, respectively. Also, it was indicated that the polysaccharide formation was significantly enhanced by adding 100 U/mL of penicillin G potassium to the cultures during the decelerating growth phase. Kurane et al. also investigated the effects of cultivation conditions on the activity of the bioflocculants, synthesized by *R. erythropolis* S-1 and 260-2 [49]. The favorable medium for production of the flocculant was defined to be 1% fructose (or glucose), 0.5% K_2_HPO_4_, 0.2% KH_2_PO_4_, 0.02% MgSO_4_, 0.01% NaCl, 0.05% urea, and 0.05% yeast extract with initial pH 9.5. In this study, EPSs were not extracted [49], as well as bioflocculant from *Rhodococcus* sp. R3 [53].

The extraction and purification procedures represent another major source of product variability. It is known that physical and chemical methods, or the combination of them, are used for EPS extraction [110]. As Table 2 shows, centrifugation cycles and ethanol precipitation are the most general methods for EPS extraction from *Rhodococcus*. However, differences in solvent concentration, temperature, and rotation speed can selectively precipitate EPS fractions of different molecular weights. The purification steps and procedures used are distinguished in terms of degree of product purity. Protein removal using trichloroacetic acid, phenol–chloroform extraction, or enzymatic digestion may further modify EPS composition, potentially removing covalently or non-covalently bound proteins and lipids. Interestingly, Hu et al., in their modified extraction method, heated the culture at 90 °C to inactivate enzymes and then centrifuged to remove cells [8]. Perry et al. centrifuged the culture at 60 °C [57]. In many studies, dialysis at 4 °C is the standard method to use for purification. In some experimental studies, only washing with ether or reprecipitation from distilled water were applied [16,56]. For QEPS, an additional centrifugation step was used to remove insoluble impurities [54].

It is difficult to estimate effects of growth conditions and extraction and purification methods on the target activities of EPSs. Not only do these parameters vary in different studies but also the species and strains used, as well as the experimental conditions, in which activities are determined. However, some conclusions are possible to be obtained from these diverse data. On average, production of purified EPSs seems to be lower than that of unpurified EPSs due to extraction and purification losses. For instance, the highest production of 10.136 g/L was achieved when a rich medium was used to grow producers, and LB-EPSs were obtained after the cell culture centrifugation and protein precipitation with TCA. The lowest EPS productions of 0.003–0.273 g/L were achieved when *Rhodococcus* cells were grown in minimal media, and LB-EPSs were obtained after centrifugation and dialysis (Table 2). The most correct comparison may be performed for flocculating activities of EPSs extracted from *R. erythropolis* ACCC 10543 [16] and *R. opacus* 89 UMCS [52]. Although representatives of various species are compared, they both have been grown in minimal media, and similar conditions (kaolin clay, CaCl_2_, EPS concentrations, and stirring time) have been used to determine flocculating rates. LB-EPSs were obtained in both cases after centrifugation, but EPS from *R. opacus* 89 UMCS was purified, and no purification was performed for EPS from *R. erythropolis* ACCC 10543 (Table 2). Similar flocculating rates of 85–88% were obtained for both EPSs that were related to a similar proportions of proteins in the EPS of 8–9% (Table 1). For flocculation, a purification step seems to be unnecessary. Probably, purified and polysaccharide-rich EPSs are required for emulsification of oils. TB-EPSs produced by *R. qingshengii* QDR4-2 and obtained after cell heating and chromatographical purification emulsified edible oils better (54–73%) [54] than unpurified LB-EPSs produced by *R. erythropolis* Au-1, which emulsified 42–58% of vaseline at similar EPS concentration [56].

Advanced purification methods, such as ion exchange chromatography and gel filtration, are effective in producing well-defined EPS fractions, but they are labor-intensive, low-throughput, and poorly suited for scale-up, which restricts their application primarily to analytical or laboratory-scale studies. Obtaining high-purity EPSs facilitates more detailed structural analyses and exploration of their significant biological activities in biomedicine. Meanwhile, in some applications (e.g., bioremediation), unpurified EPSs can be highly efficient [56,59]. For example, Shi et al. obtained a protein-enriched fraction of extracellular outer matter through optimized incubation and filtration, since proteins were shown to be key contributors to the resuscitation-promoting activity [59]. This optimized sample markedly promoted resuscitation of nonculturable cells and significantly enhanced PCB degradation. Influence of purification, as well as effects of growth conditions, on heavy metal sorption, hydrocarbon biodegradation, and emulsification by EPSs isolated from *Rhodococcus* cells are not clear. Excluding these two cases [56,59], EPSs purified with dialysis or chromatography methods (Table 2) were used for bioremediation purposes [33,36,46,47,52]. To be honest, the LB-EPSs and minimal salt media supplemented with applied growth substrates was less labor-intensive.

Overall, the data summarized in Table 2 emphasize that EPS production by *Rhodococcus* is condition-dependent, with cultivation and extraction parameters exerting a certain effect on polymer composition and functionality. These factors, combined with challenges in scale-up and downstream processing, highlight the need for standardized growth strategies, scalable extraction methods, and comprehensive structure–function analyses.

## 6. Conclusions and Future Challenges

This article provides a detailed review of the chemical composition, biological functions, and activities of EPSs produced by actinomycetes of the genus *Rhodococcus*, discovered over recent decades. EPSs from *Rhodococcus* represent a big and diverse class of biologically derived polymers, distinguished by their structural diversity, surface activity, and provided properties. As this review illustrates, *Rhodococcus* EPSs have already demonstrated substantial value in their capacities for hydrocarbon emulsification, flocculation, ion sorption, and biofilm formation, providing powerful tools for bioremediation. Importantly, growing evidence now supports their emerging relevance in biomedical sciences. As discussed above, several independent studies have explored the potential activities of *Rhodococcus* exopolysaccharides, including antioxidant, antiviral, anticancer, antibiofilm, anti-inflammatory, and viscosity properties. Inherent biocompatibility, non-toxicity, and tunable chemical composition make *Rhodococcus* EPSs promising candidates for next-generation biomaterials and active agents for biomedicine, environmental, food, and other industries. Despite the extensive research into *Rhodococcus* EPS activities, there are still no patented products in any of the promising areas of EPS application. Therefore, fundamental and applied investigations remain significant in a broad field of study.

Purified exopolysaccharides provide activities, making them ideal candidates for drug delivery systems and anticancer treatment procedures [6,8]. Moreover, unpurified exopolymers, remaining less studied, could extend valuable properties and application possibilities. According to structural and physiological features of EPSs described in the literature, it should be noted that, for example, lipid-rich exopolymers with small amounts of proteins and nucleic acids [5,35] could have promising potentials, especially as prebiotics. Also, bacteria have an effect on pathological formation of minerals such as gallstones and kidney stones [111], and since EPSs from *Rhodococcus* were shown to influence the mineralization process [105], this line of biomedicine research could be quite valuable.

Looking forward, expanding the biomedical potential of *Rhodococcus* EPSs will require deeper mechanistic investigations into structure–function relationships, systematic methods for scalable and controlled biosynthesis, and advanced chemical or genetic engineering to tailor molecular architecture. Integrating omics technologies, synthetic biology, and polymer engineering strategies may enable the rational design of EPS-based biopolymers with precise physicochemical and biological properties. Interestingly, genetic and metabolic engineering have not been extensively used to enhance production of EPSs in *Rhodococcus*. In only one work [81], the gene *gmhD* (D-glycero-D-manno-heptose 1-phosphate guanosyltransferase), which is important in the biosynthesis of lipopolysaccharide, was deleted to verify the electrotransformation system used for the gene knockout and to determine whether the electrotransformation efficiency was seriously affected by the presence of abundant EPSs in *R. ruber* YYL. By bridging environmental biotechnology and biomedical polymer science, *Rhodococcus* EPSs hold promise not only for sustainable remediation but also for innovative therapeutic and biomedical material development. A broad spectrum of in vivo tests is required to be performed to define therapeutic effects and a biosafety level for *Rhodococcus* EPSs to apply them in medicine.

Future investigations of EPSs produced by *Rhodococcus* bacteria will be also related to the analysis of less studied species. Currently, a total of 59 valid species of the genus *Rhodococcus* are known (https://lpsn.dsmz.de/genus/rhodococcus, last accessed 17 December 2025). Available publications on *Rhodococcus* EPSs are restricted to several species. They are devoted mainly to *R. erythropolis* and *R. rhodochrous*; less often to *R. equi*, *R. pyridinivorans* and *R. opacus*; and a few works are devoted to other species. As shown in our recent work [5], these less-studied species can be promising producers of EPSs. The six diverse genes encoding glycosyltransferases have been found in *Rhodococcus cerastii* and *Rhodococcus corynebacterioides* strains that can be a basis for unique and specific chemical compositions of EPSs. The glycosyltransferase potentially responsible for enhanced synthesis of EPSs have been found in *R. ruber* strains [5]. Moreover, EPSs produced by *R. coprophilus* are not studied. This species is a known bioindicator for fecal contamination [112]. In one available work [42], EPSs of *R. coprophilus* were shown to bind human viruses, which could be important for epidemiological investigations.

Variations of the chemical composition and structure of EPSs from *Rhodococcus* caused by substrates and growth conditions are important to guarantee stable properties of these compounds in time. Apparently, suitability and stability of EPSs will depend on the form or type of the EPS-based biopreparation. Extracted EPSs seem to be stable, although this stability can be temporary and limited to the time of life of the EPSs. They can be used as emulsifying agents, flocculants, or sorbents for heavy metals and toxic organic compounds. Extracted EPSs will be degraded with time. We assume that time of degradation will depend on molecular weight, length of chains, and chemical composition. However, biodegradability seems to be a useful property. Biodegradable compounds and materials are environmentally friendly and do not contaminate the environment. If separation of EPS-bound contaminants will be a next step, when EPSs are applied, their time of life is not really significant. For long-term applications, this parameter should be preliminarily estimated. In case viable *Rhodococcus* cells producing EPSs during biotechnological processes are used, stability of synthesizing EPSs cannot be completely guaranteed. Thus, mechanisms of EPS biosynthesis by a target strain (strains) and sensitivity of EPS biosynthesis to changes in the cell surrounding must be preliminarily investigated and then monitored.

Careful attention is required with respect to methods for extraction and purification of EPSs produced by *Rhodococcus* bacteria. We cannot recommend developing universal standards for these procedures since they will depend on a combination of factors, including cost efficiency, labor intensity, their feasibility and suitability, as well as purposes, achievement of high target activities, appropriate biosafety parameters, and compatibility with the environment. Standard procedures to grow *Rhodococcus* cells and isolate and purify EPSs may be required to compare and achieve the same activities in various EPSs. It is recommended for researchers to keep in mind and pay attention to extraction and purification protocols for future reproducibility of properties for obtained EPSs.

## Figures and Tables

**Figure 1 ijms-27-00498-f001:**
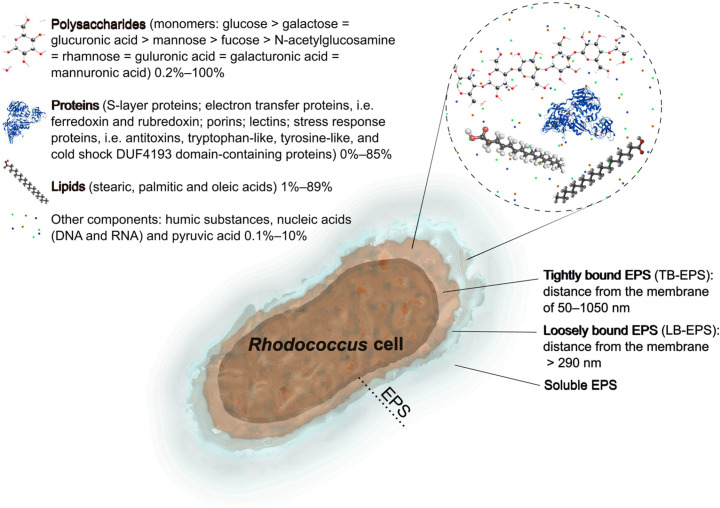
Fractions and summarized composition of EPSs produced by *Rhodococcus* bacteria. Monomers of polysaccharides are listed according to their occurrence in EPSs.

**Figure 2 ijms-27-00498-f002:**
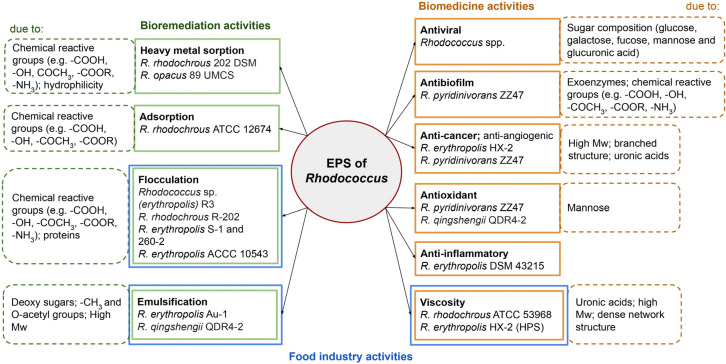
The activities of *Rhodococcus* EPSs and their causes, with examples of strains showing these activities. Mw—molecular weight.

## Data Availability

Data are contained within the article.

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
