# Peer review of "Extracellular Polymeric Substances Produced by Actinomycetes of the Genus Rhodococcus for Biomedical and Environmental Applications"

_ijms, 2026, doi:10.3390/ijms27010498_

Round 1
Reviewer 1 Report
Comments and Suggestions for Authors
This is a well written review on the genus Rhodococcus.
As it covers environmental aspects perhaps R. coprophilus as bioindicator organism of the farm pollutants might be added. Examples include: Rowbotham, T. J., & Cross, T. (1977). Ecology of Rhodococcus coprophilus and associated actinomycetes in fresh water and agricultural habitats. Microbiology, 100(2), 231-240.; Mara, D. Duncan, and John I. Oragui. "Occurrence of Rhodococcus coprophilus and associated actinomycetes in feces, sewage, and freshwater." Applied and environmental microbiology 42, no. 6 (1981): 1037-1042; Wicki, Melanie, Adrian Auckenthaler, Richard Felleisen, Marianne Liniger, Caroline Loutre, Isabel Niederhauser, Marcel Tanner, and Andreas Baumgartner. "Improved detection of Rhodococcus coprophilus with a new quantitative PCR assay." Applied microbiology and biotechnology 93, no. 5 (2012): 2161-2169.
Key words might also contain words related to environment.
Rhodococcus actinomycetes perhaps can be changed into Rhodococcus genus of the class actinomycetes.
Author Response
Responses to the Reviewer 1
Comment. This is a well written review on the genus Rhodococcus.
Answer. Thank you for a high estimation of our work.
Comment. As it covers environmental aspects perhaps R. coprophilus as bioindicator organism of the farm pollutants might be added. Examples include: Rowbotham, T. J., & Cross, T. (1977). Ecology of Rhodococcus coprophilus and associated actinomycetes in fresh water and agricultural habitats. Microbiology, 100(2), 231-240.; Mara, D. Duncan, and John I. Oragui. "Occurrence of Rhodococcus coprophilus and associated actinomycetes in feces, sewage, and freshwater." Applied and environmental microbiology 42, no. 6 (1981): 1037-1042; Wicki, Melanie, Adrian Auckenthaler, Richard Felleisen, Marianne Liniger, Caroline Loutre, Isabel Niederhauser, Marcel Tanner, and Andreas Baumgartner. "Improved detection of Rhodococcus coprophilus with a new quantitative PCR assay." Applied microbiology and biotechnology 93, no. 5 (2012): 2161-2169.
Answer. Thank you for the valuable comment. R. coprophilus is a known bioindicator for faecal contamination. Unfortunately, there is no available publications with detailed description of EPS produced by R. coprophilus in big literature search systems (Scopus, PubMed, Google Academia, Consensus). In one work (Santiso-Bellón et al. Rhodococcus spp. interacts with human norovirus in clinical samples and impairs its replication on human intestinal enteroids. Gut Microbes, 2025, 17 (1), 2469716. https://doi.org/10.1080/19490976.2025.2469716), it was found that EPS of R. coprophilus bound human viruses, as well as EPS of some other Rhodococcus species did the same. R. coprophilus is not only one poorly studied Rhodococcus species in terms of their EPS. There are other ecologically and biotechnologically important species (R. jostii, R. ruber), which EPS are also poorly studied. We have added R. coprophilus in Conclusion as an important species for future research (p. 23, lines 818-821, a clean version).
Comment. Key words might also contain words related to environment.
Answer. The key word “environmental protection” has been added to summarize all possible applications of Rhodococcus EPS in this area, including not only bioremediation but also wastewater treatment and neutralization of contaminants in contained systems.
Comment. Rhodococcus actinomycetes perhaps can be changed into Rhodococcus genus of the class actinomycetes.
Answer. Thank you for the point on correct taxonomy and spelling. We have changed the name “Rhodococcus actinomycetes” on “actinomycetes of the genus Rhodococcus” in title, keywords, abstract (p. 1, lines 10-11, a clean version) and main text (p. 3, line 92; p. 10, line 312; p. 22, line 768, a clean version). This is more precise than our original variant, a little bit shorter than you suggest, highlight a group of the bacteria studied (class Actinomycetes) among all microorganisms, and specify target microorganisms. In the text, we also use word combinations “Rhodococcus bacteria”, “Rhodococcus cells”, “Rhodococcus species”, rhodococci, and sometimes (in long sentences) “Rhodococcus actinomycetes”.
Reviewer 2 Report
Comments and Suggestions for Authors
This manuscript provides a detailed description of the structure and technological applications of Rhodococcus EPS. Overall, the authors have carefully reviewed the development and applications of these exopolysaccharides, and the writing is generally clear. However, the authors must address the following issues; otherwise, the manuscript cannot be considered for further publication.
1.As stated in line 40, EPS production is common among many microorganisms. The reason why the authors specifically selected Rhodococcus remains unclear. Please clarify this in the Introduction for the benefit of readers. Why not choose E. coli, a facultative and fast-growing bacterium? A discussion of E. coli–derived exopolymers may help strengthen the rationale. This reference may be facilitated to address this concern: 10.1021/acsestengg.3c00472.
2.In line 14, the description of the role of EPS applies to all microorganisms, not only Rhodococcus.
3.In line 17, the manuscript mentions “unique structures” and “changes,” but does not explain what these structures are or what changes occur. Please provide concrete and informative details.
4.In line 22, the authors emphasize the structural diversity of EPS. Can this structural diversity guarantee the suitability and stability of EPS in environmental applications—for example, bioremediation? The reviewer recommends elaborating or clarifying this point.
5.In line 82, regarding the classification of EPS: I understand the TB-EPS and other categories, but could such classification be further defined based on the distance from the cell membrane, rather than simply describing EPS as “tightly bound” or “loosely bound”?
6.In line 123, although the authors present the diversity of EPS, it is unclear whether this diversity is due to intrinsic microbial characteristics or is regulated by growth conditions. If the latter, the reviewer recommends supplementing the table with information on the growth conditions under which different EPS are synthesized.
7.Most importantly, although the manuscript reviews many studies emphasizing the environmental applications of EPS, it does not clarify whether the EPS are isolated and then applied (e.g., for heavy-metal adsorption). The manuscript also lacks a description and summary of EPS isolation or extraction techniques. If the intention is only to rely on Rhodococcus-produced EPS for heavy-metal removal, the toxic effects of heavy metals on the cells themselves must also be considered. Overall, this point represents a major issue that must be addressed before the manuscript can be accepted.
Author Response
Responses to the Reviewer 2
This manuscript provides a detailed description of the structure and technological applications of Rhodococcus EPS. Overall, the authors have carefully reviewed the development and applications of these exopolysaccharides, and the writing is generally clear. However, the authors must address the following issues; otherwise, the manuscript cannot be considered for further publication.
Answer. Thank you for a high estimation of our work and a careful revision. Your valuable comments help to improve text and make it more attractive, useful and interesting for readers.
- As stated in line 40, EPS production is common among many microorganisms. The reason why the authors specifically selected Rhodococcus remains unclear. Please clarify this in the Introduction for the benefit of readers. Why not choose E. coli, a facultative and fast-growing bacterium? A discussion of E. coli–derived exopolymers may help strengthen the rationale. This reference may be facilitated to address this concern: 10.1021/acsestengg.3c00472.
Answer. This is a very reasonable criticism. It is completely truth that fast-growing and well-known microorganisms, like Escherichia coli, Pseudomonas putida and aeruginosa, Bacillus subtilis, are perfect biotechnology agents and producers. However, not only these well-studied bacteria are used for synthesis of EPS. Xanthomonas produce xanthan, and some specific bacteria produce bacterial cellulose, which have unique properties. Thus, search for novel producers can be promising. Extremotolerant bacteria-degraders of xenobiotics and producers of various second metabolites like non-pathogenic actinomycetes of some specific genera (Rhodococcus are promising among them because of their catabolic redundance) are less sensitive to changes in growth conditions, better survive and are better stored, can be used in the couple with waste conversion technologies, and are suitable for clean-up and remediation. Moreover, EPS produced by Rhodococcus are not properly studied, and both their common and specific characteristics in comparison with EPS from other microorganisms are not clear.
We have added two sentences in Introduction to stress on these properties of Rhodococcus, highlight their biotechnological advantages and contour specific areas of application in comparison with E. coli (p. 2, lines 75-81, a clean version).
- In line 14, the description of the role of EPS applies to all microorganisms, not only Rhodococcus.
Answer. The sentence has been changed and extended to show the same set of biological functions like in other bacteria and highlight participation of EPS in resistance of rhodococci against toxic compounds and assimilation of hydrophobic substances. These functions of EPS are frequently mentioned for xenobiotic degraders.
The current version of the sentence (p. 1, lines 15-20, a clean version): “Rhodococcus species synthesize complex EPS composed primarily of polysaccharides, proteins and lipids that, like in other bacteria, support cell adhesion, aggregation, biofilm formation, and horizontal gene transfer (as well as can prevent exogenous DNA binding), and are highly important for resistance against toxicants and dissolution / assimilation of hydrophobic compounds.”
- In line 17, the manuscript mentions “unique structures” and “changes,” but does not explain what these structures are or what changes occur. Please provide concrete and informative details.
Answer. The sentence has been changed, details have been added, and “unique structures” have been deleted (a comprehensive comparison with other microorganisms is required to postulate them unique). The current version of the sentence is (p. 1, lines 20-25, a clean version):
“EPS produced by different species of Rhodococcus exhibit diverse structures (soluble EPS, loosely bound and tightly bound fractions, capsules, linear and branched chains, amorphous coils, rigid helices, mushroom-like structures, extracellular matrix, and a fibrillar structure with a sheet-like texture), leading to variations in their properties (rheological features, viscosity, flocculation, sorption abilities, compression, DNA binding, and interaction with hydrophobic substrates).”
- In line 22, the authors emphasize the structural diversity of EPS. Can this structural diversity guarantee the suitability and stability of EPS in environmental applications—for example, bioremediation? The reviewer recommends elaborating or clarifying this point.
Answer. This is a very good question. Thanks to the reviewer. This issue, indeed, should be addressed.
Suitability and stability of EPS produced by Rhodococcus cells will depend on the form or type of the biopreparation. Extracted EPS, apparently, seem to be stable in their chemical composition and structure, at least temporary stable. Extracted EPS can be used as emulsifying agents, flocculants, or sorbents for heavy metals and toxic organic compounds. Of course, EPS will be degraded with time. We assume that time of degradation will depend on molecular weight, length of chains and chemical composition of EPS. Biodegradability is a useful property of chemicals and preparations. Biodegradable compounds and materials are considered to be environmentally friendly, not contaminating the environment. If separation of bound contaminants is a next step in application of EPS, their time of life is not really significant. In other cases, it should be preliminary estimated and taken into account.
In case viable Rhodococcus cells producing EPS during biodegradation processes are used, stable properties of synthesized EPS cannot be completely guaranteed. Thus, mechanisms of EPS biosynthesis by a target strain (strains) and sensitivity of EPS biosynthesis to changes in the cell surrounding must be preliminary investigated and then monitored.
A paragraph discussing these possible problems in application of EPS from Rhodococcus for the environmental protection is added to the text (p. 23-24, lines 822-838, a clean version).
- In line 82, regarding the classification of EPS: I understand the TB-EPS and other categories, but could such classification be further defined based on the distance from the cell membrane, rather than simply describing EPS as “tightly bound” or “loosely bound”?
Answer. The information on distances for various fractions of EPS has been added (p. 3, lines 104-116, a clean version):
EPS make up a significant part of the biopolymers synthesized by cells. The thickness of EPS surrounding cells can be similar or even larger the cell diameter. According to measurements performed with an atomic-force microscope, the thickness of TB-EPS produced by Rhodococcus cells varies between 240 nm and 1,000 nm depending on the density of the EPS inner layer (Pen et al. Effect of extracellular polymeric substances on the mechanical properties of Rhodococcus. BBA – Biomembranes, 2015, 1848(2), 518–526, https://doi.org/10.1016/j.bbamem.2014.11.007). Summarizing with average thickness of the call wall of gram-positive bacteria being ~50 nm (Mitchell et al. Critical cell wall hole size for lysis in Gram-positive bacteria. Journal of the Royal Society Interface, 2013, 10(80), 20120892, https://doi.org/10.1098/rsif.2012.0892; Mai-Prochnow et al. Gram positive and Gram negative bacteria differ in their sensitivity to cold plasma. Scientific Reports, 2016, 6, 38610, https://doi.org/10.1038/srep38610), TB-EPS are located on the distance of 50–1,050 nm from the cytoplasmic membrane. LB-EPS are more diffuse, form the outer EPS layer, are weakly bound with the inner layer and, apparently, located on the distance > 290 nm from the membrane (thickness of the cell wall 50 nm + minimal thickness of the EPS inner layer 240 nm). Interestingly, that capsules, which also consist of extracellularly located polysaccharides, can reach the similar thickness, from 100 nm until 2,500 nm, as shown, for example, for clinical isolates of Streptococcus pneumonia (Eichner et al. Intra-serotype variation of Streptococcus pneumoniae capsule and its quantification. Microbiology Spectrum, 2025, 13, e03087-24, https://doi.org/10.1128/spectrum.03087-24).
- In line 123, although the authors present the diversity of EPS, it is unclear whether this diversity is due to intrinsic microbial characteristics or is regulated by growth conditions. If the latter, the reviewer recommends supplementing the table with information on the growth conditions under which different EPS are synthesized.
Answer. This is a very good point, which must be addressed. Apparently, growth conditions impact on the composition and properties of EPS produced by Rhodococcus bacteria. There are not many examples described in the literature, which directly compare EPS synthesized under different growth conditions (sodium acetate and dibenzofuran, as an example). Actually, researchers use various, probably, feasible and convenient in their studies growth conditions, various strains and species, and whole picture of how differences in cultivation conditions effect on EPS is difficult to be obtained. Also, the opinion exists that growth conditions do not really influence on the EPS composition and properties. The Table 2 has been created, where growth conditions, methods for extraction and purification of EPS, and their influence on the EPS production by Rhodococcus cells are presented. Influence of growth conditions is also described in the additional chapter created – subsection 5: p. 17-22, lines 656-765, a clean version.
- Most importantly, although the manuscript reviews many studies emphasizing the environmental applications of EPS, it does not clarify whether the EPS are isolated and then applied (e.g., for heavy-metal adsorption). The manuscript also lacks a description and summary of EPS isolation or extraction techniques. If the intention is only to rely on Rhodococcus-produced EPS for heavy-metal removal, the toxic effects of heavy metals on the cells themselves must also be considered. Overall, this point represents a major issue that must be addressed before the manuscript can be accepted.
Answer. Thank you for these very important comments. Examples, where EPS of Rhodococcus have been applied for heavy metal sorption, include growth of Rhodococcus cells in a minimal salt medium with glucose, obtaining loosely bound EPS fraction by centrifugation and its purification using filtration and dialysis (Czemierska et al. Production and characterisation of exopolymer from Rhodococcus opacus. Biochemical Engineering Journal, 2016, 112, 143–152, https://doi.org/10.1016/j.bej.2016.04.015; Dobrowolski et al. Studies of cadmium(II), lead(II), nickel(II), cobalt(II) and chromium(VI) sorption on extracellular polymeric substances produced by Rhodococcus opacus and Rhodococcus rhodochrous. Bioresource Technology, 2017, 225, 113–120, https://doi.org/10.1016/j.biortech.2016.11.040; Dobrowolski et al. Extracellular polymeric substances immobilized on microspheres for removal of heavy metals from aqueous environment. Biochemical Engineering Journal, 2019, 143, 202–211, https://doi.org/10.1016/j.bej.2019.01.004). This procedure is reasonable, since exopolysaccharide functional groups bind ions, especially cations. However, it is not clear, if unpurified EPS, which are more cost-efficient, have same activity. This issue must be addressed in future investigations for the real biotechnology sector. For example, Shi et al. (Extracellular organic matter-mediated self-regulation of indigenous Rhodococcus sp. enhances PCB biodegradation under environmental stress: Self-recovery strategy for sustained bioremediation. Environmental Research, 2025, 285, 122716, https://doi.org/10.1016/j.envres.2025.122716) used almost unpurified EPS produced by Rhodococcus sp. SJ for the polychlorbiphenyls’ degradation. As well as, the growth medium optimization should be also addressed. This part of the discussion has been added to the chapter 5 “Influence of growth conditions and extraction methods on composition and properties of Rhodococcus EPS” and chapter 6 “Conclusion and future challenges” – p. 22, lines 744-759, a clean version, and p. 23-24, lines 822-845, a clean version, respectively.
Concerning toxicity of heavy metals, soluble (extracted) EPS were applied in the literature (Czemierska et al., 2016, https://doi.org/10.1016/j.bej.2016.04.015; Dobrowolski et al., 2017, https://doi.org/10.1016/j.biortech.2016.11.040; Dobrowolski et al., 2019, https://doi.org/10.1016/j.bej.2019.01.004). However, rhodococci are known to be highly tolerant to heavy metal salts (Ivshina et al. Adaptive mechanisms of nospecific resistance to heavy metal ions in alkanotrophic actinobacteria. Russian Journal of Ecology, 2013, 44(1), 123–130, https://doi.org/10.1134/S1067413613020082; Ivshina et al. Bioaccumulation of molybdate ions by alkanotrophic Rhodococcus leads to significant alterations in cellular ultrastructure and physiology. Ecotoxicology and Environmental Safety, 2024, 274, 116190, https://doi.org/10.1016/j.ecoenv.2024.116190; Golysheva et al. Diversity of mercury-tolerant microorganisms. Microorganisms, 2025, 13(6), 1350, https://doi.org/10.3390/microorganisms13061350) with resistance to ≥ 200 mg/L, and concentrations of 5–700 mg/L used in works (Czemierska et al., 2016, https://doi.org/10.1016/j.bej.2016.04.015; Dobrowolski et al., 2017, https://doi.org/10.1016/j.biortech.2016.11.040; Dobrowolski et al., 2019, https://doi.org/10.1016/j.bej.2019.01.004) seem not to be a restriction – p. 15, lines 555-560, a clean version.
Reviewer 3 Report
Comments and Suggestions for Authors
The manuscript provides a detailed review of extracellular polymeric substances (EPS) produced by Rhodococcus actinomycetes, summarizing their chemical composition, biological functions, and applications in biomedicine, bioremediation, and other fields. The work consolidates a broad range of studies and emphasizes the biotechnological potential of these biopolymers. However, the review lacks critical analysis in certain areas, and modifications are needed to improve its scholarly rigor and comprehensiveness. The specific points for revision are as follows:
- The review frequently cites "unpublished data" (e.g., in discussions of Rhodococcus EPS composition and activities), which undermines the credibility and reproducibility of the content. How can the authors replace these references with peer-reviewed publications to strengthen the evidence base?
- Figure 1 and Figure 2 are schematic diagrams that oversimplify the complexity of EPS fractions and activities. How can the authors enhance these figures with more detailed labels, quantitative data, or real-world examples to better support the textual descriptions?
- The discussion of biomedical applications (e.g., antiviral, anticancer effects) relies heavily on in vitro studies without addressing clinical translatability or mechanistic depth. How can the authors incorporate limitations, such as the lack of in vivo validation, and propose future research directions to bridge this gap?
- The literature coverage appears uneven, with emphasis on specific Rhodococcus species (e.g., R. erythropolis) while neglecting others. How can the authors ensure a balanced inclusion of recent studies (post-2023) and underrepresented species to avoid bias and reflect the full diversity of EPS producers?
- The organization of sections (e.g., chemical composition vs. biological functions) leads to redundancy, such as repeated mentions of EPS roles in biofilm formation. How can the authors restructure the content to improve flow, eliminate repetition, and enhance logical progression?
- The conclusion section identifies general future prospects but fails to outline specific, actionable research priorities (e.g., standardizing EPS extraction methods or exploring genetic engineering). How can the authors provide concrete recommendations to guide future studies effectively?
- The review extensively covers EPS properties but neglects to discuss the methodological inconsistencies in EPS extraction and purification across cited studies, which may lead to biased comparisons of composition and bioactivities. How can the authors add a subsection evaluating common extraction techniques (e.g., chemical vs. physical methods) and propose guidelines for standardizing protocols to improve reproducibility in future research?
- While emphasizing the benefits of Rhodococcus EPS in bioremediation, the manuscript omits any discussion of potential ecological risks, such as unintended microbial community shifts or persistence of EPS by-products in environments. How can the authors address these limitations by incorporating case studies on environmental monitoring or regulatory frameworks to provide a more balanced perspective on safe application?
Author Response
Responses to the Reviewer 3
The manuscript provides a detailed review of extracellular polymeric substances (EPS) produced by Rhodococcus actinomycetes, summarizing their chemical composition, biological functions, and applications in biomedicine, bioremediation, and other fields. The work consolidates a broad range of studies and emphasizes the biotechnological potential of these biopolymers. However, the review lacks critical analysis in certain areas, and modifications are needed to improve its scholarly rigor and comprehensiveness.
Answer. Thank you for a high estimation of our work, careful revision and valuable comments to improve the quality of the manuscript. We have tried to answer all the issues raised.
The specific points for revision are as follows:
- The review frequently cites "unpublished data" (e.g., in discussions of Rhodococcus EPS composition and activities), which undermines the credibility and reproducibility of the content. How can the authors replace these references with peer-reviewed publications to strengthen the evidence base?
Answer. These are not unpublished data actually. They are mentioned by authors in original research papers as unpublished. We have changed subsequent sentences and deleted the word combination “unpublished data”:
- 9, line 288, a clean version;
- 13, lines 447-448, a clean version;
- 14, line 500, a clean version.
2. Figure 1 and Figure 2 are schematic diagrams that oversimplify the complexity of EPS fractions and activities. How can the authors enhance these figures with more detailed labels, quantitative data, or real-world examples to better support the textual descriptions?
Answer. The figures have been modified. More details including specification of chemical composition of EPS components, the distance of various fractions of EPS from the membrane, wider ranges of percentages of specific components, and less schematic visualization of various classes of biopolymers as structures in the Figure 1, as well as, specification of functional groups in the Figure 2. We added more information in the figures; they looked like a plain text oversaturated with information. Our aim is to summarize chemical composition of EPS produced by Rhodococcus cells in the Figure 1 and applications because of activities, which in their turn are due to chemical composition and properties, of EPS in the Figure 2 schematically, in broad strokes.
3. The discussion of biomedical applications (e.g., antiviral, anticancer effects) relies heavily on in vitro studies without addressing clinical translatability or mechanistic depth. How can the authors incorporate limitations, such as the lack of in vivo validation, and propose future research directions to bridge this gap?
Answer. Thank you for this right question. Yes, biological activities of many substances produced by living organisms and intended to be applied in medicine as drugs are frequently determined in vitro. This is not bad actually and, anyway, adds to the pool of potentially promising pharmaceuticals. But this lack of information exists and should be addressed. EPS produced by Rhodococcus bacteria are not an exception. Almost all their biological activities are defined in vitro. Only in one study (TaÅŸkaya et al. 2023, https://doi.org/10.56430/japro.1307611), they were checked in in vivo systems (chicken eggs and mice). Human beings have never been used in these studies. We have added the discussion on this gap in approvement of biological activities of Rhodococcus EPS to the text (p. 13, lines 453-465, a clean version):
It is important to note that most studies of biological activities of EPS produced by Rhodococcus bacteria were performed in vitro. Only in one work, anti-angiogenic effects of EPS produced by R. pyridinivorans ZZ47 were estimated on embryos in fertilized chicken eggs, as well as a single dose acute toxicity test was conducted using laboratory mice (TaÅŸkaya et al. Exopolysaccharide from Rhodococcus pyridinivorans ZZ47 strain: Evaluation of biological activity and toxicity. Journal of Agricultural Production, 2023, https://doi.org/10.56430/japro.1307611). According to this single study, EPS of Rhodococcus are not toxic and can be used to prevent a blood supply of tumors. On the other side, a revealed dose-dependent anti-angiogenic activity is a concern to use the ZZ47 EPS in pregnant women (because of possible adverse effects on embryos) or at regeneration of tissues; and their local application can be a better option. However, there is a lack of sufficient information to declare therapeutic effects and biosafety of EPS produced by Rhodococcus cells that can be a direction for future investigations. A deep purification of EPS with controlled chemical composition is required to apply these compounds in medicine that must be taken into account at development of EPS-based pharmaceuticals.
As well as recommendation to perform in vivo studies in future has been added to Conclusions: p. 23, lines 805-807.
4. The literature coverage appears uneven, with emphasis on specific Rhodococcus species (e.g., R. erythropolis) while neglecting others. How can the authors ensure a balanced inclusion of recent studies (post-2023) and underrepresented species to avoid bias and reflect the full diversity of EPS producers?
Answer. This is another reasonable and good question. However, this is not our choice and selection. There is a shifted focus of publications on several species. We have looked carefully through publications for all years (including 2025) using various search engines (PubMed, Scopus, Google Academy, Consensus) and key words “Rhodococcus”, “EPS”, “extracellular polymeric substances”, “exopolysaccharides”, and “exopolymers”. In this review, all found publications are mentioned. Of course, we do not pretend that completely all papers are covered but focus on certain species and unrepresentation of other species is a tendency. Currently, the total 59 valid species of the genus Rhodococcus are known (https://lpsn.dsmz.de/genus/rhodococcus). Available publications on Rhodococcus EPS are devoted mainly to R. erythropolis / R. qingshengii (this is one species https://lpsn.dsmz.de/species/rhodococcus-qingshengii) and R. rhodochrous, less – to R. equi, R. pyridinivorans and R. opacus, and a few works are devoted to other species. As shown in our recent work (Krivoruchko et al. The lipid- and polysaccharide-rich extracellular polymeric substances of Rhodococcus support biofilm formation and protection from toxic hydrocarbons. Polymers, 2025, 17(14), 1912, https://doi.org/10.3390/polym17141912), these less studied species can be promising producers of EPS. The 6 diverse genes encoded glycosyltransferases have been found in Rhodococcus cerastii and Rhodococcus corynebacterioides strains that can be a basis for unique and specific chemical composition of their EPS. The glycosyltransferase potentially responsible for enhanced synthesis of EPS has been found in R. ruber strains. This discussion has been added to the text – p. 23, lines 808-821, a clean version.
5. The organization of sections (e.g., chemical composition vs. biological functions) leads to redundancy, such as repeated mentions of EPS roles in biofilm formation. How can the authors restructure the content to improve flow, eliminate repetition, and enhance logical progression?
Answer. We have reduced the chapter 3 “Biological functions” to eliminate repetitive information of the EPS influence on the biofilm formation and to have a focus on confirmation of protective functions of EPS produced by Rhodococcus bacteria. Additionally, we have changed the title of the chapter 2 as “Chemical composition and its influence on activities and functions of EPS” to highlight the relationship between chemical components and biological activities + functions of these Rhodococcus substances.
6. The conclusion section identifies general future prospects but fails to outline specific, actionable research priorities (e.g., standardizing EPS extraction methods or exploring genetic engineering). How can the authors provide concrete recommendations to guide future studies effectively?
Answer. Thank you for this valuable comment. Phrases and paragraphs focusing on these issues have been added: p. 21-22, lines729-765; p. 23, lines 797-803; and p. 24, lines 839-845, a clean version. It is interesting to note that genetic and metabolic engineering have not extensively used to enhance production of EPS in Rhodococcus. Only in one work (Huang et al. Efficient electrotransformation of Rhodococcus ruber YYL with abundant extracellular polymeric substances via a cell wall-weakening strategy. FEMS Microbiol Lett., 2021, 368(9), fnab049, https://doi.org/10.1093/femsle/fnab049), the gene gmhD (D-glycero-D-manno-heptose 1-phosphate guanosyltransferase), which is important in the biosynthesis of lipopolysaccharide, was deleted to verify the electrotransformation system used for the gene knockout and to determine whether the electrotransformation efficiency was seriously affected by the presence of abundant EPS in R. ruber YYL.
7. The review extensively covers EPS properties but neglects to discuss the methodological inconsistencies in EPS extraction and purification across cited studies, which may lead to biased comparisons of composition and bioactivities. How can the authors add a subsection evaluating common extraction techniques (e.g., chemical vs. physical methods) and propose guidelines for standardizing protocols to improve reproducibility in future research?
Answer. The subsection 5 “Influence of growth conditions and extraction methods on composition and properties of Rhodococcus EPS” (p. 22, lines 744-759) has been added. Pshrase with a focus on reproducibility of results because of differences in EPS extraction and purification methods has been added too – p. 24, lines 839-845, a clean version.
8. While emphasizing the benefits of Rhodococcus EPS in bioremediation, the manuscript omits any discussion of potential ecological risks, such as unintended microbial community shifts or persistence of EPS by-products in environments. How can the authors address these limitations by incorporating case studies on environmental monitoring or regulatory frameworks to provide a more balanced perspective on safe application?
Answer. This is a very good and reasonable question, and an uncovered gap in our manuscript. Two paragraphs to discuss ecological risks of Rhodococcus EPS have been added – p. 16, lines 585-623, a clean version:
At bioremediation, especially in open and fragile ecosystems, evaluation of ecological risks related with bioremediation agents is essential. Although, all EPS are mixtures of biopolymers, which can be degraded by microorganisms, their fate in the environment, time of life and impacts on ecosystems can result in significant effects. As efficient sorbents and traps for metal ions and organic molecules, they can adsorb and retain dangerous toxicants acting as a sink and secondary contamination. It was shown for lake-derived EPS of sludge, which tightly bound perfluorooctanoic acid and perfluorooctane sulfonate. Stable self-assembled complexes were formed, altering EPS conformation and protein secondary structure. Binding with N-containing compounds (EPS proteins and amino sugars) resulted in formation of compact multilayered shells and persistence of these fluoroalkyl substances. Phytotoxicity tests indicated great environmental risks of these complexes and their inhibiting effects on plants (Zhang et al. Binding stability of per- and polyfluoroalkyl substances to sludge extracellular polymeric substances drives increased environmental risks. Journal of Hazardous Materials, 2025, 496, 139320, https://doi.org/10.1016/j.jhazmat.2025.139320). EPS used separately from producing them Rhodococcus biodegraders can prevent biodegradation of contaminants, binding and holding their molecules. It was shown for nonylphenol, which resulted in formation of aromatic, humified and hydrophobic tyrosine-like, tryptophan-like, and soluble microbial by-product-like substances in colloidal EPS. These complexes were toxic for sedimental bacteria, and degradation of nonylphenol was reduced. Addition of Fe2O3 nanoparticles softened these negative effects (Cheng et al. Effects of Fe2O3 nanoparticles on extracellular polymeric substances and nonylphenol degradation in river sediment. Science of the Total Environment, 2021, 770, 145210, https://doi.org/10.1016/j.scitotenv.2021.145210). On the other hand, binding of toxicants reduces their migration and acute toxicity to living organisms. It should be taken into account that application of EPS alone or bound with live Rhodococcus cells for sorption and chelating heavy metals does not provide elimination of these emergent pollutants. After degradation of EPS and death of Rhodococcus producers, heavy metal ions release into the environment again. Development of technologies when Rhodococcus EPS with accumulated heavy metals can be replaced and processed further ex situ can help to overcome this inconvenience.
Another concern is possible antimicrobial activities of Rhodococcus EPS applied in large amounts, that can affect indigenous microorganism. In the literature, these activities of microbial EPS are considered to be beneficial and prevent food spoilage and biofouling (Guezennec et al. Exopolysaccharides from unusual marine environments inhibit early stages of biofouling. International Biodeterioration & Biodegradation, 2012, 66(1), 1–7. https://doi.org/10.1016/j.ibiod.2011.10.004; Abdalla et al. Exopolysaccharides as antimicrobial agents: Mechanism and spectrum of activity. Frontiers in Microbiology, 2021, 12, 664395. https://doi.org/10.3389/fmicb.2021.664395). In our work (Krivoruchko et al., 2025, https://doi.org/10.3390/polym17141912), presence of a big proportion of lipids in LB-EPS produced by Rhodococcus spp. can supply microorganisms with additional available nutrients, promote growth of satellite species and stimulate formation of a microbial food web. It is recommended to estimate complex ecological effects of Rhodococcus EPS before their application. The total ecological risk will also depend on the form of EPS (soluble or cell-bound) and a system (open or contained) where they are applied. Additionally, pathogenicity of strains must be addressed in case viable Rhodococcus producers are introduced. Species of the genus Rhodococcus belong to the risk group 1 (https://lpsn.dsmz.de/genus/rhodococcus, last accessed 24 December 2025), excluding one opportunistic species R. equi (https://lpsn.dsmz.de/species/rhodococcus-equi, last accessed 24 December 2025) and a phytopathogenic species R. fascians (Stamler et al. First report of Rhodococcus isolates causing pistachio bushy top syndrome on ‘UCB-1’ rootstock in California and Arizona. Plant Disease, 2015, 99(11), 1468–1476, https://doi.org/10.1094/PDIS-12-14-1340-RE).
Round 2
Reviewer 2 Report
Comments and Suggestions for Authors
I have carefully reviewed the authors’ point-by-point responses to the reviewers’ comments, as well as the corresponding revisions made to the manuscript. I must say that this is one of the most thorough and conscientious revisions I have encountered in recent submissions. After careful re-evaluation, I believe that the manuscript has adequately addressed all comments and is now suitable to proceed to the next stage of publication. I am confident that the publication of this work will make a valuable contribution to the application of EPS produced by Rhodococcus actinomycetes. Congratulations to all the authors.
Author Response
Thank you very much for a high estimation of our work. It was a pleasure to analyze literature additionally more critically and deeper after your comments. They stimulated us to add essential details, which, no doubts, made the review more comprehensive and interesting for readers. Thank you again for a careful attention to our work.
Reviewer 3 Report
Comments and Suggestions for Authors The authors have diligently addressed several reviewer comments in their revised manuscript, two minor issues require further attention to meet International Journal of Molecular Sciences's rigorous standards for the quality of papers. 1. While the review comprehensively summarizes the diverse structures and bioactivities of Rhodococcus EPS, the discussion in Sections 4.1 and 4.2 could be strengthened by providing a more detailed mechanistic explanation of how specific structural features (e.g., monosaccharide composition, glycosidic linkages, or the presence of fatty acid esters and uronic acids detailed in Sections 2.1 and 2.3) directly translate into the observed biological functions, such as antiviral activity or heavy metal biosorption. A deeper exploration of the structure-activity relationships, perhaps by correlating specific chemical motifs (like the 5-amino-3,5-dideoxynonulosonic acid in R. equi serotype 4 or the fatty acid content in EPS from R. erythropolis PR4) with their respective mechanisms of action (e.g., molecular mimicry in virus binding or hydrophobic interactions in emulsification) would significantly enhance the mechanistic insight and provide a clearer roadmap for future material design based on Rhodococcus EPS. 2. The manuscript highlights in Section 5 that cultivation conditions and extraction methods significantly influence EPS yield, composition, and properties, as summarized in Table 2. However, the narrative would benefit from a more critical discussion on how these methodological variations specifically impact the reproducibility and cross-comparison of functional data (e.g., emulsification indices, flocculating rates, or antioxidant activities) reported across different studies. Elaborating on the potential for standardizing key parameters (such as carbon source selection or purification techniques) for specific application targets (biomedical vs. environmental) would provide valuable guidance for ensuring consistent and reproducible outcomes in future research and potential industrial applications, thereby addressing a critical aspect of data reliability in this field.Author Response
Responses to the Reviewer 3
The authors have diligently addressed several reviewer comments in their revised manuscript, two minor issues require further attention to meet International Journal of Molecular Sciences's rigorous standards for the quality of papers.
- While the review comprehensively summarizes the diverse structures and bioactivities of Rhodococcus EPS, the discussion in Sections 4.1 and 4.2 could be strengthened by providing a more detailed mechanistic explanation of how specific structural features (e.g., monosaccharide composition, glycosidic linkages, or the presence of fatty acid esters and uronic acids detailed in Sections 2.1 and 2.3) directly translate into the observed biological functions, such as antiviral activity or heavy metal biosorption. A deeper exploration of the structure-activity relationships, perhaps by correlating specific chemical motifs (like the 5-amino-3,5-dideoxynonulosonic acid in R. equi serotype 4 or the fatty acid content in EPS from R. erythropolis PR4) with their respective mechanisms of action (e.g., molecular mimicry in virus binding or hydrophobic interactions in emulsification) would significantly enhance the mechanistic insight and provide a clearer roadmap for future material design based on Rhodococcus EPS.
Answer. Thank you for this question. Yes, we agree that mechanisms must be designated more clearly to enhance scientific soundness of this review. We apologize that we did not properly and correctly estimate the significance and depth of this question in the first revision, this is our fault.
EPS produced by Rhodococcus spp. in the work Santiso-Bellón et al. (Rhodococcus spp. interacts with human norovirus in clinical samples and impairs its replication on human intestinal enteroids. Gut Microbes, 2025, https://doi.org/10.1080/19490976.2025.2469716) contain histo-blood groups antigens (HBGA), which a typically recognized by noroviruses. This explanation is presented in the text (p. 12, lines 384-402, a clean version). Additionally, presence of HBGA motifs in EPS produced by Rhodococcus strains in this work, has been added into Table 1; and discussion on significance not only spatial structure but flexibility of epitopes on the virus binding has been added (p. 12, lines 394-400, a clean version).
EPS produced by R. qingshengii QDR4-2 and R. pyridinivorans ZZ47 show antioxidant activities. These activities can be related to mannose or uronic acid residues. This discussion has been added to the text (p. 12, lines 409-420, a clean version).
Anti-angiogenic activity of the EPS produced by R. pyridinivorans ZZ47 may be closely linked to their antioxidant activity. Microbial EPS with antioxidant properties inhibit tumor angiogenesis related genes expressions and upregulate anti-angiogenic genes in some cancer cell lines that can be related to the abruption of signal transduction pathways. Antibiofilm activities of EPS can be caused by degrading extracellular enzymes incorporated in the EPS matrix or inhibiting effects of EPS on the bacterial cell adhesion due to the presence of negatively charged functional groups (Figure 2). Concerning antibiofilm activities of rhodococcal EPS, it was shown that the adhesion force of the cantilever of an atomic-force microscope to crude EPS produced by various Rhodococcus species was on 1–3 nN lower than to unmodified cover glass (Krivoruchko et al. The lipid- and polysaccharide-rich extracellular polymeric substances of Rhodococcus support biofilm formation and protection from toxic hydrocarbons. Polymers 2025, https://doi.org/10.3390/polym17141912). Although, no correlation was found between the EPS productions and adhesive activities of studied Rhodococcus strains to polystyrene, but these EPS could strongly affect the adhesion of other bacteria. Adhesion is the first step of the biofilm formation, and inhibition of this process results in stopping or reduced growth of microbial biofilms. This paragraph has been added to the text on p. 13, lines 433-448, a clean version.
Likely, selective and dose-dependent cytotoxic activities of rhodococcal EPS, namely HPS produced by R. erythropolis HX-2, against tumor cell lines and their neutral effects on normal cells (fibroblasts L929 in work of Hu et al. Purification, characterization and anticancer activities of exopolysaccharide produced by Rhodococcus erythropolis HX-2. International Journal of Biological Macromolecules 2020, https://doi.org/10.1016/j.ijbiomac.2019.12.228) are related to specific ligand-lectin interactions. Tumor cells, for example, can express galactins (galactose specific receptors) and mannose-receptors. Galactose and mannose are in the composition of HPS (Table 1). Moreover, HPS proteins can participate in specific EPS-cell interactions. These interactions can be a trigger for the programmed cell death or inhibit cell proliferation. This text has been added in p. 13, lines 458-464, a clean version.
Some EPS, like the extracellular heteropolysaccharide produced by R. erythropolis DSM 43215, show anti-inflammatory effects. They can have complex mechanisms and multiple causes, such as specific interaction of oligosaccharide motifs and proteins of EPS with mannose-like, Toll-like and other receptors on endothelial or immune cells, interaction with components of glycocalyx resulting in their binding (cationic inflammatory mediators, cytokines, compliment, etc.) or changes in charge and viscosity, binding with lipopolysaccharides of pathogenic bacteria, chelating metal cations, and involvement in reduction of oxidative stress, as discussed above. Lipids in Rhodococcus EPS can potentially provide inflammatory activities through binding prostaglandins. This part of the discussion has been added in p. 13, lines 468-477, a clean version.
In p. 14, lines 482-496, a clean version, details of antigen structures for capsular polysaccharides of the animal pathogen R. equi have been added. Their variety could be important to obtain effective vaccines.
Capabilities of Rhodococcus EPS, such as gelling, water-absorbing and water-holding activities, and their application as thickeners, are related with their 3D-porous structure, not with stereospecific molecular interactions and chemical composition of monomers in polysaccharide chains. This is discussed in p. 14, lines 497-515, a clean version. A short discussion on influence of other components (lipids, pyruvic acid) on these properties of EPS has been added in lines 516-521.
Less information has been added to the chapter 4.2 Bioremediation: p. 15, lines 554-557; p. 16, lines 610-620; and p. 17, lines 659-665, a clean version. Participation of EPS in degradation of hydrophobic pollutants is related to protective function of EPS (barrier and buffer zone + stress resistance proteins), electron transfer to oxygenases, and optimal hydrophobicity/hydrophilicity to contact with hydrophobic substrate. Flocculating activities of EPS depend on protein amounts. Uronic acids are important to attract metal cations. Density of EPS (density of functional groups finally) is important for flocculation and ion sorption. Lipids, apparently, are key components in emulsifying activities of EPS produced by Rhodococcus bacteria.
- The manuscript highlights in Section 5 that cultivation conditions and extraction methods significantly influence EPS yield, composition, and properties, as summarized in Table 2. However, the narrative would benefit from a more critical discussion on how these methodological variations specifically impact the reproducibility and cross-comparison of functional data (e.g., emulsification indices, flocculating rates, or antioxidant activities) reported across different studies. Elaborating on the potential for standardizing key parameters (such as carbon source selection or purification techniques) for specific application targets (biomedical vs. environmental) would provide valuable guidance for ensuring consistent and reproducible outcomes in future research and potential industrial applications, thereby addressing a critical aspect of data reliability in this field.
Answer. Thank you for this valuable comment. It is difficult to estimate effects of growth conditions, extraction and purification methods on target activities of EPS. Not only these parameters vary in different studies but also the species and strains used, as well as experimental conditions, in which activities are determined. However, some conclusions are possible to be obtained from these diverse data. In average, production of purified EPS seems to be lower than that of unpurified EPS due to extraction and purification losses. For instance, the highest production of 10.136 g/L was achieved when a rich medium was used to grow producers, and LB-EPS were obtained after the cell culture centrifugation and protein precipitation with trichloroacetic acid. The lowest EPS productions of 0.003–0.273 g/L were achieved when Rhodococcus cells were grown in minimal media, and LB-EPS were obtained after centrifugation and dialysis (Table 2). The most correct comparison may be done for flocculating activities of EPS extracted from R. erythropolis ACCC 10543 (Peng et al. Characterization and application of bioflocculant prepared by Rhodococcus erythropolis using sludge and livestock wastewater as cheap culture media, Appl Microbiol Biotechnol 2014, https://doi.org/10.1007/s00253-014-5725-4) and R. opacus 89 UMCS (Czemierska et al. Production and characterisation of exopolymer from Rhodococcus opacus. Biochem Eng J 2016, https://doi.org/10.1016/j.bej.2016.04.015). Although, representatives of various species are compared, they both have been grown in minimal media, and similar conditions (kaolin clay, CaCl2, EPS concentrations and stirring time) have been used to determine flocculating rates. LB-EPS were obtained in both cases after centrifugation, but EPS from R. opacus 89 UMCS was purified, and no purification was performed for EPS from R. erythropolis ACCC 10543 (Table 2). Similar flocculating rates as 85%–88% were obtained for both EPS that was related to similar proportions of proteins in the EPS as 8%–9% (Table 1). For flocculation, a purification step seems to be unnecessary. Probably, purified and polysaccharide-rich EPS are required for emulsification of oils. TB-EPS produced by R. qingshengii QDR4-2 and obtained after cell heating and chromatographical purification emulsified edible oils better (54%–73%) (Li et al. Purification, structural characterization, antioxidant and emulsifying capabilities of exopolysaccharide produced by Rhodococcus qingshengii QDR4-2. J Polym Environ 2023, https://doi.org/10.1007/s10924-022-02604-0) than unpurified LB-EPS produced by R. erythropolis Au-1, which emulsified 42%–58% of vaseline at similar EPS concentration (Semeniuk et al. Biosynthesis and characteristics of metabolites of Rhodococcus erythropolis Au-1 strain. J Microbiol Biotechnol Food Sci 2022, https://doi.org/10.55251/jmbfs.4714). This paragraph has been added in p. 24, lines 833-856, a clean version; as well as a conclusion has been modified – p. 26, lines 957-958, a clean version.